# The ubiquitin-conjugating enzyme UBE2K determines neurogenic potential through histone H3 in human embryonic stem cells

Azra Fatima[1], Dilber Irmak[1], Alireza Noormohammadi[1], Markus M. Rinschen [1], Aniruddha Das [2], Orsolya Leidecker[3], Christina Schindler[1], Víctor Sánchez-Gaya[4], Prerana Wagle[1], Wojciech Pokrzywa [1,2], Thorsten Hoppe[1,5], Alvaro Rada-Iglesias[4,5] & David Vilchez [1,5 ✉]

Histones modulate gene expression by chromatin compaction, regulating numerous processes such as differentiation. However, the mechanisms underlying histone degradation remain elusive. Human embryonic stem cells (hESCs) have a unique chromatin architecture characterized by low levels of trimethylated histone H3 at lysine 9 (H3K9me3), a heterochromatin-associated modification. Here we assess the link between the intrinsic epigenetic landscape and ubiquitin-proteasome system of hESCs. We find that hESCs exhibit high expression of the ubiquitin-conjugating enzyme UBE2K. Loss of UBE2K upregulates the trimethyltransferase SETDB1, resulting in H3K9 trimethylation and repression of neurogenic genes during differentiation. Besides H3K9 trimethylation, UBE2K binds histone H3 to induce its polyubiquitination and degradation by the proteasome. Notably, *ubc-20*, the worm orthologue of UBE2K, also regulates histone H3 levels and H3K9 trimethylation in *Caenorhabditis elegans* germ cells. Thus, our results indicate that UBE2K crosses evolutionary boundaries to promote histone H3 degradation and reduce H3K9me3 repressive marks in immortal cells.

[1] Cologne Excellence Cluster for Cellular Stress Responses in Aging-Associated Diseases (CECAD), University of Cologne, Joseph Stelzmann Strasse 26, 50931 Cologne, Germany. [2] Laboratory of Protein Metabolism in Development and Aging, International Institute of Molecular and Cell Biology, Trojdena Street 4, 02-109 Warsaw, Poland. [3] Max Planck Institute for Biology of Ageing, Joseph-Stelzmann-Strasse 9b, Cologne 50931, Germany. [4] Institute of Biomedicine and Biotechnology of Cantabria (IBBTEC), University of Cantabria, Albert Einstein 22, 39011 Santander, Spain. [5] Center for Molecular Medicine Cologne (CMMC), University of Cologne, Robert Koch Strasse 21, 50931 Cologne, Germany. ✉email: dvilchez@uni-koeln.de

Embryonic stem cells (ESCs) can replicate indefinitely while retaining their potential to differentiate into all cell lineages[1,2]. A precisely coordinated network of transcriptional and epigenetic modifiers determines ESC identity. For instance, ESCs exhibit an endogenous transcriptional network that modulates their self-renewal and pluripotency, including transcription factors such as OCT4, NANOG and SOX2[3]. Likewise, chromatin modifiers regulate ESC pluripotency and differentiation[4,5]. In comparison with their differentiated counterparts, ESCs have a unique chromatin architecture such as fewer condensed heterochromatin foci, a pattern that facilitates dynamic reorganization of chromatin during development[6]. For instance, heterochromatin-associated histone modifications such as H3K9 trimethylation (H3K9me3) are usually reduced in ESCs[6–8]. Notably, regulators of histone modifications and chromatin compaction are required for ESC differentiation, including the polycomb repressive complex 1 (PRC1)[9,10]. In these lines, huntingtin protein (HTT) binds and inhibits SETDB1, a methyltransferase that specifically trimethylates lysine 9 of histone H3. Concomitantly, HTT maintains low levels of H3K9me3 in human ESCs (hESCs)[11,12].

Although histones are key determinants of chromatin compaction, little is known about how their total levels are modulated. The ubiquitin-proteasome system is involved in the regulation of histone levels[13], but the enzymes that catalyze their polyubiquitination for proteasome recognition have not been identified in multicellular organisms[14]. The attachment of ubiquitin is achieved through a sequential mechanism involving three classes of enzymes[15]. First, the ubiquitin-activating enzyme (E1) activates the C-terminal glycine residue of ubiquitin in an ATP-dependent manner. Activated ubiquitin is next transferred to a ubiquitin-conjugating enzyme (E2). In the third step, a ubiquitin ligase (E3) attaches ubiquitin from the E2 enzyme to the target protein. The same cascade can link additional molecules to the primary ubiquitin via internal ubiquitin lysines, forming a ubiquitin chain. Ubiquitination can affect numerous proteins in many manners: it can signal for their degradation through the proteasome or modulate their activity, intracellular localization, and interaction with other proteins. E2 enzymes are the main determinants for selection of the lysine to construct ubiquitin chains, which thereby control the cellular fate of the substrate. In humans, 35 E2 and over 600 E3 enzymes have been identified so far. Growing evidence indicates that ESCs not only exhibit increased proteasome activity, but also an intrinsic network of ubiquitin ligases[16–22]. As such, the ubiquitin-proteasome system has a central role in the immortality and cell fate decisions of pluripotent stem cells[16–22].

Given their endogenous chromatin structure signature, ESCs could provide a novel paradigm to discover epigenetic regulatory mechanisms and their impact on differentiation. For this purpose, we ask whether the intrinsic ubiquitin-proteasome system of hESCs impinge on their epigenetic landscape. Notably, we find that the ubiquitin-conjugating enzyme E2 K (UBE2K), also known as huntingtin-interacting protein 2 (HIP2), is upregulated in hESCs compared with their differentiated counterparts. UBE2K reduces total histone H3 levels and trimethylation of H3K9, allowing for neurogenesis of hESCs. However, loss of UBE2K impairs the levels of SETDB1 and its regulator HTT, resulting in H3K9 trimethylation and repression of neurogenic genes during differentiation. Besides H3K9me3 regulation, UBE2K binds histone H3 and promotes its degradation primarily by 26S proteasomes, regulating total H3 protein amounts. Interestingly, UBE2K also impinges upon total H3 and H3K9me3 levels in the organismal model *Caenorhabditis elegans*, particularly in the immortal germline. Thus, our results provide a link between the ubiquitin-proteasome system and histone regulation in both ESCs and germ cells.

## Results

**hESCs exhibit increased levels of UBE2K.** To determine changes in the levels of E2 enzymes during differentiation, we analyzed available quantitative proteomics data[23]. We found 4 E2 enzymes (i.e., UBE2C, UBE2G1, UBE2K and UBE2O) increased in hESCs when compared with their neural progenitor cell (NPC) and neuronal counterparts (Supplementary Table 1). Notably, UBE2K markedly decreased during neural differentiation of hESCs (Supplementary Table 1), as we confirmed by western blot analysis (Fig. 1a). In both NPCs and neurons, the decrease in the protein amount of UBE2K correlated with a reduction of the transcript levels (Fig. 1b). UBE2K protein amounts also decreased during differentiation into either endoderm or mesoderm, indicating that this is not a specific phenomenon associated to neural differentiation (Fig. 1c, d). We then asked whether high levels of UBE2K can be reprogrammed. Indeed, induced pluripotent stem cells (iPSCs) displayed increased levels of UBE2K compared with their parental fibroblasts (Fig. 1e, f). Taken together, our results indicate that enhanced UBE2K protein expression is associated with pluripotency.

**Loss of UBE2K impairs neurogenesis from hESCs.** With the strong correlation between UBE2K expression and pluripotency, we assessed whether increased levels of UBE2K are required to maintain the undifferentiated state of hESCs. For this purpose, we generated stable UBE2K knockdown hESC lines using two independent shRNAs. Since we did not observe morphological differences, we analysed their transcriptome (Supplementary Data 1). We found a statistically significant >2-fold-change in only 35 transcripts, of which 12 were downregulated whereas 23 were upregulated (Fig. 2a and Supplementary Data 1). Among the changed transcripts, gene ontology biological process (GOBP) term analysis indicated enrichment for modulators of transcription (Fig. 2b and Supplementary Data 1). To further identify changes in UBE2K shRNA hESCs, we performed quantitative proteomics (Supplementary Data 2). Besides UBE2K, these experiments revealed that other 67 proteins are decreased in both independent UBE2K shRNA hESC lines (Supplementary Data 3). Among them, GOBP analysis revealed the strongest enrichment for genes involved in translational regulation (Fig. 2c and Supplementary Data 3). In addition, loss of UBE2K resulted in the upregulation of 46 proteins which were enriched for transcriptional modulators (Fig. 2c and Supplementary Data 4). Despite these moderate changes in the proteome, loss of UBE2K did not impair the expression of pluripotency markers (Fig. 2d, e). Since hESC lines can vary in their characteristics, we examined an independent line and obtained similar results (Supplementary Fig. 1). Moreover, knockdown of UBE2K did not change the expression of distinct germ layers markers (Fig. 2f). Thus, our results indicate that loss of UBE2K does not induce differentiation of hESCs.

Although UBE2K knockdown did not significantly alter the levels of pluripotency markers in hESCs, another possibility is that UBE2K determines their ability to differentiate. Given that the ubiquitin-proteasome system is required for neurogenesis from hESCs[20–22], we focused on the neural lineage. For this purpose, we performed neural induction and monitored the expression of PAX6, an early marker of neuroectodermal differentiation[24]. After 10 days on neural induction treatment, control hESCs differentiated into early NPCs that express high levels of PAX6 (Fig. 3a–c). Similarly, PAX6 expression was triggered to the same extent in UBE2K shRNA hESCs after 10 days of neural induction, resulting in cultures essentially formed by PAX6-positive cells (Fig. 3a–c). Besides PAX6, we assessed the levels of other neural markers (Nestin, SOX1) and

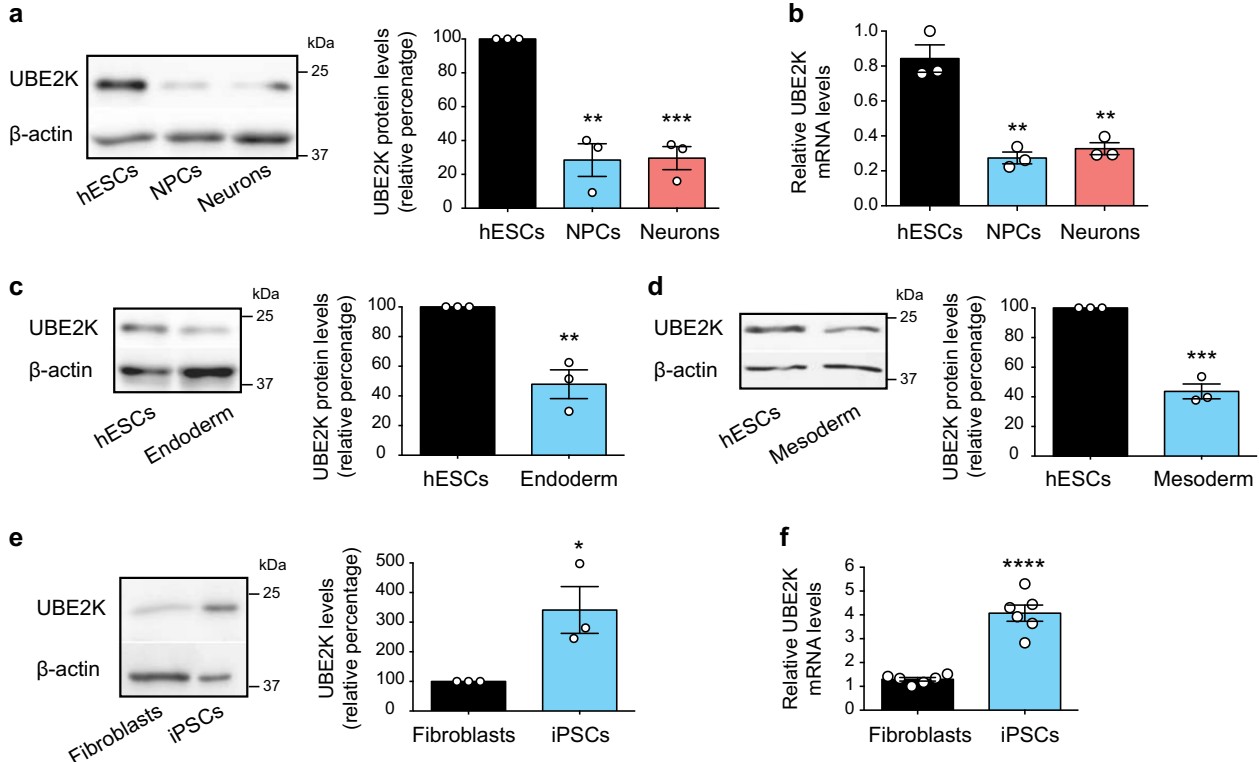

**Fig. 1 The expression of UBE2K decreases during differentiation. a** Western blot analysis with antibody to UBE2K. The graph represents the UBE2K relative percentage values (corrected for β-actin loading control) to H9 hESCs (mean ± s.e.m. of three independent experiments). **b** Quantitative PCR (qPCR) analysis of UBE2K mRNA levels. Graph (relative expression to H9 hESCs) represents the mean ± s.e.m. of three biologically independent samples. **c, d** Western blot analysis with antibody to UBE2K. The graphs represent the UBE2K relative percentage values (corrected for β-actin) to H9 hESCs (mean ± s.e.m. of three independent experiments). **e** Western blot analysis of UBE2K levels comparing iPSCs with their parental HFF-1 fibroblasts. The graph represents the UBE2K relative percentage values (corrected for β-actin) to HFF-1 fibroblasts (mean ± s.e.m. of three independent experiments). **f** UBE2K mRNA relative levels to HFF1 fibroblasts represent the mean ± s.e.m. ($n = 6$ biologically independent samples). All the statistical comparisons were made by two-tailed Student's $t$-test for unpaired samples. $P$ value: $*P < 0.05$, $**P < 0.01$, $***P < 0.001$, $****P < 0.0001$.

found no significant differences in their induction at the early NPC stage (Fig. 3c and Supplementary Fig. 2). When compared with hESCs, early NPCs also had higher expression of MAP2, a microtubule-associated protein which is upregulated through the differentiation process into neurons[25,26]. However, loss of UBE2K did not affect MAP2 induction during the first 10 days of neural differentiation (Fig. 3c). Likewise, knockdown of UBE2K did not impair the induction of neural markers and MAP2 in a distinct hESC line during the early stages of neural differentiation (Supplementary Fig. 3). Thus, these results suggest that UBE2K is not required for the commitment of hESCs to a neuroectoderm fate. However, we observed dramatic differences when we further differentiated these cells (20 days on neural induction) to obtain mature NPCs with the ability to generate terminally differentiated neurons (Fig. 3d–g). At this stage, the expression of the early NPC marker PAX6 decreases while the levels of MAP2 are upregulated[25]. Indeed, we observed that control mature NPCs lose the expression of PAX6, but exhibit increased levels of other neural markers (Nestin and SOX1) as well as neuronal factors (e.g., MAP2, NEFL, AADC, SYN1, GABAR) when compared with early NPCs (Fig. 3f, g). On the contrary, UBE2K shRNA cells retained abnormal high levels of PAX6 whereas the expression of Nestin, SOX1 and most of the neuronal markers tested was lower compared with control mature NPCs in two independent cell lines (Fig. 3d–g and Supplementary Fig. 4a, b). Therefore, knockdown of UBE2K in hESCs could impair their ability to generate mature NPCs. In contrast to control mature NPCs, UBE2K shRNA cells did not develop neuronal extensions even

after terminal neuronal differentiation treatment during one month (Fig. 3h and Supplementary Fig. 5a). Moreover, these cells exhibited a strong impairment in the neuronal induction of MAP2, neurofilaments and synaptic proteins while retaining high levels of PAX6 expression (Fig. 3h–k and Supplementary Fig. 5b). Taken together, these data suggest a role of UBE2K in maintaining the ability of hESCs to differentiate into mature NPCs with intact neurogenic properties.

**Loss of UBE2K increases H3 levels and H3K9me3 in hESCs.** UBE2K interacts with huntingtin protein (HTT)[27]. Our recent findings indicate that HTT maintains low levels of H3K9 tri-methylation in hESCs[12]. Importantly, H3K9 trimethylation induced by HTT knockdown does not alter the undifferentiated state of hESCs and their commitment into early NPCs[12]. However, it particularly impairs the transition of early NPCs into mature NPCs and neuronal differentiation[12]. Given that loss of UBE2K induces a similar phenotype, we asked whether UBE2K is also involved in histone H3 regulation. Although highly abundant in the cytoplasm, UBE2K was also present in nuclear fractions of hESCs, supporting a potential role in the nucleus (Supplementary Fig. 6). Notably, co-immunoprecipitation experiments followed by western blot indicated that UBE2K interacts with histone H3 (Fig. 4a). In addition, loss of UBE2K was sufficient to increase the total protein levels of histone H3 in hESCs (Fig. 4b). On the contrary, we did not observe interaction of UBE2K with histone H1 or changes in H1 protein levels upon UBE2K shRNA

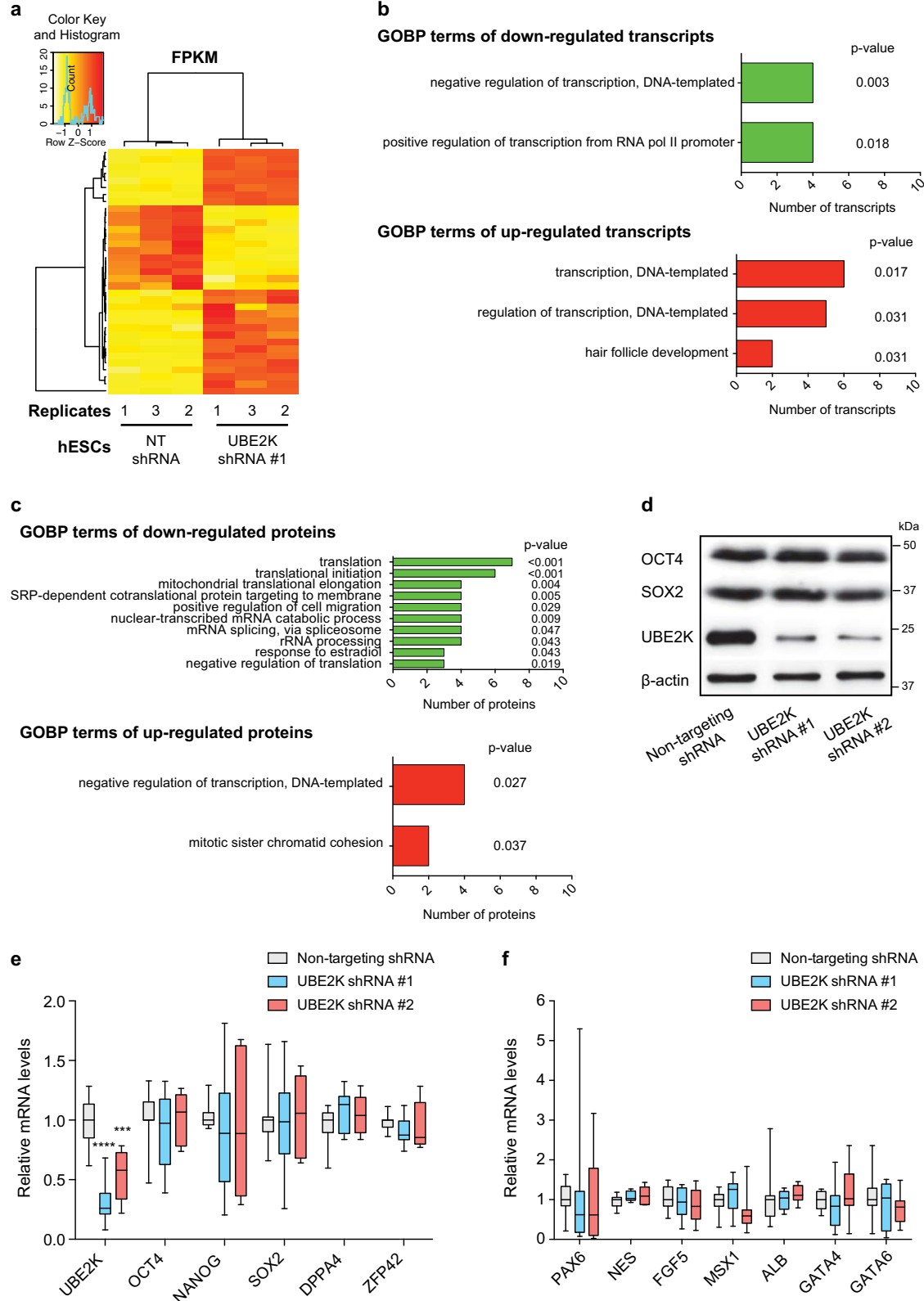

(Fig. 4a, b). Since UBE2K knockdown did not affect the transcripts amounts of H3 (Fig. 4c), our results suggest a role of UBE2K in the regulation of H3 protein levels. Besides total H3, UBE2K knockdown also increased global H3K9me3 levels (Fig. 4d, e). More importantly, we found enrichment for H3K9me3 fraction among total histone H3 (Fig. 4d). Likewise, UBE2K shRNA upregulated both H3 levels and H3K9me3/H3

ratio in an independent hESC line (Supplementary Fig. 7). Collectively, our results suggest that endogenous high expression of UBE2K not only reduces total histone H3 but also H3K9 trimethylation in hESCs.

Given the robust increase in H3K9me3/H3 ratio induced by loss of UBE2K, we asked whether UBE2K also modulates other H3 modifications. Upon UBE2K knockdown, H3K9me1/H3 and

**Fig. 2 Knockdown of UBE2K does not affect the expression of pluripotency markers. a** Heatmap representing differentially expressed transcripts identified by RNA-sequencing analysis (Fold-change > 2, $P$ value < 0.05, $n = 3$ biologically independent samples) in UBE2K shRNA H9 hESCs compared with non-targeting (NT) shRNA hESCs. **b** GOBP analysis of downregulated and upregulated transcripts in UBE2K shRNA hESCs. Statistical comparisons with a modified Fisher's exact test (EASE score) below the $P$ value cutoff of 0.05 were considered significant. **c** GOBP analysis of downregulated and upregulated proteins in both UBE2K shRNA #1 and shRNA #2 H9 hESCs. For downregulated proteins, 10 of the 23 enriched GOBPs are shown. Please see Supplementary Data 3 for a complete list of enriched GOBP terms. $P$ value (EASE score) < 0.05 was considered significant. **d** Western blot analysis of H9 hESCs with antibodies to OCT4, SOX2 and UBE2K. β-actin is the loading control. Images are representative of three independent experiments. **e** qPCR analysis of UBE2K and pluripotency markers in H9 hESCs. Graph (relative expression to NT shRNA control hESCs) represents the mean ± s.e.m. of nine independent experiments. **f** qPCR analysis of ectodermal (PAX6, NES, FGF5), mesodermal (MSX1) and endodermal (ALB, GATA4, GATA6) germ layer markers. Graph (relative expression to NT shRNA) represents the mean ± s.e.m. of nine independent experiments. In (**e**, **f**) statistical comparisons were made by two-tailed Student's $t$ test for unpaired samples. $P$ values: ***$P$ < 0.001, ****$P$ < 0.0001.

H3K9me2/H3 ratios were diminished and not significantly changed, respectively (Fig. 5a). Whereas loss of UBE2K also induced an upregulation in the global levels of other H3 modifications (i.e., H3K4me3, H3K27me3, H3K27ac), the ratio of these modifications was not significantly enriched when normalized by total H3 amounts (Fig. 5b). In addition, co-immunoprecipitation experiments indicate a more robust interaction of UBE2K with H3K9me3 than H3K4me3, H3K27me3, and H3K27ac (Fig. 5c). Thus, UBE2K could particularly modulate H3K9 trimethylation. Since HTT inhibits the H3K9 methyltransferase activity of SETDB1 in hESCs[12], we first confirmed that UBE2K interacts with HTT in these cells (Fig. 5d). Notably, loss of UBE2K resulted in a downregulation of HTT levels, correlating with higher H3K9 trimethylation (Fig. 5e). Besides HTT, we found that UBE2K also interacts with SETDB1 (Fig. 5d). Moreover, knockdown of UBE2K dramatically increased the amounts of SETDB1 (Fig. 5e). Concomitantly, histone H3 gained interaction with SETDB1 in hESCs, a process that could contribute to H3K9 trimethylation on UBE2K knockdown (Fig. 5f).

**H3K9me3 represses the induction of distinct neural genes.** Since UBE2K decreases during differentiation, we asked whether this downregulation correlates with changes in histone H3. Whereas the levels of H3K9me3 increased upon neural differentiation of control hESCs, we did not observe changes in total H3 levels (Supplementary Fig. 8), indicating that other mechanisms could contribute to maintaining physiological amounts of histone H3. However, when UBE2K shRNA hESCs were differentiated into early NPCs, these cells retained the alterations in both total H3 and H3K9 trimethylation induced at the undifferentiated state (Fig. 6a, b and Supplementary Fig. 9). The increase in total H3 levels of NPCs derived from UBE2K shRNA hESCs was not associated with changes in the amounts of histone H3 transcripts (Supplementary Fig. 10a, b). Altogether, these data suggest that epigenetic changes induced by UBE2K knockdown in hESCs are transmitted to their neural counterparts.

To assess whether H3K9 trimethylation induced by UBE2K knockdown in hESCs affects gene expression, we first performed chromatin immunoprecipitation-sequencing (ChIP-seq) using an antibody to H3K9me3 (Supplementary Data 5). We found $a > 1.5$ fold-change enrichment ($P$ value < 0.05) for H3K9me3 marks in 821 gene-associated regions upon UBE2K shRNA in hESCs (Fig. 7a and Supplementary Data 5). Among them, we found factors involved in transcriptional regulation such as several zinc finger proteins (e.g., ZNF236, ZNF416, ZNF844, ZNF879), the transcription factor GBX1 and the bHLH transcription cofactor HES6 (Supplementary Data 5). GBX1 is highly expressed in the neuroectoderm and modulates midbrain/forebrain formation by determining the positioning of the midbrain-hindbrain boundary organizer in the early neural plate[28]. HES6 promotes neuronal differentiation by allowing the transcription factor ASCL1 to

induce the expression of genes required for neurogenesis at early stages of development[29]. Besides neurogenic transcription factors, loss of UBE2K also induced an enrichment for H3K9me3 marks in other genes involved in nervous system formation (e.g., CELSR1, CPEB1[30], SMC1B). In addition, we found H3K9me3 enrichment in distinct potassium voltage-gated channels (KCNA1, KCNA3, KCNA5, KCNF1) (Supplementary Data 5). Thus, we asked whether H3K9me3 modifications induced by UBE2K knockdown result in decreased expression of these pro-neuronal factors. However, we did not find significant changes in their expression at the hESC stage (Fig. 7b and Supplementary Fig. 11a). In addition, early NPCs from UBE2K shRNA hESCs (10 days of neural induction) did not exhibit significant decreased expression of H3K9me3-enriched genes when compared with control NPCs (Fig. 7c). Since these genes were typically expressed at low amounts in control hESCs and early NPCs but induced during neuronal differentiation (Fig. 7d and Supplementary Fig. 11b), we asked whether abnormal H3K9me3 marks impair their induction. Indeed, knockdown of UBE2K in hESCs diminished their ability to trigger the expression of distinct pro-neuronal genes (e.g., HES6, KCNA3, KCNA5) upon further neural induction (20 days) and terminal neuronal differentiation (Fig. 7e, f). Likewise, we obtained similar results in an independent hESC line (Supplementary Fig. 11c, d). To further examine the link between UBE2K, H3K9me3 and induction of neuronal genes, we performed rescue experiments. For this purpose, we knocked down UBE2K using a shRNA against the 3′-UTR region of UBE2K transcripts and rescue H3K9me3 levels by ectopic expression of UBE2K cDNA in hESCs (Fig. 8a). Upon 20 days on neural induction treatment, 3′-UTR UBE2K shRNA cells exhibited decreased expression of H3K9me3-marked genes (i.e., HES6, GBX1, KCNA3, KCNA5) when compared with control cells (Fig. 8b). Notably, ectopic expression of UBE2K not only rescued this phenotype (Fig. 8b) but also the low expression of other neural and neuronal markers such as MAP2 (Fig. 8c). Taken together, our data indicate that UBE2K maintains low levels of H3K9me3 marks in distinct genes, allowing neuronal differentiation of hESCs. Conversely, loss of UBE2K dysregulates H3K9me3 in hESCs, a process that could contribute to altered gene expression upon differentiation and compromised neurogenesis.

**UBE2K modulates ubiquitination and degradation of H3.** Besides H3K9 trimethylation, our results indicate that UBE2K also regulates total H3 proteins levels (Fig. 4b). By high-coverage sequencing of purified histones based on filter-aided sample preparation (FASP)[31], we found two ubiquitination sites at Lys18 and Lys56 of histone H3 (Supplementary Fig. 12). Since UBE2K interacts with histone H3 (Fig. 4a), we assessed whether UBE2K modulates its ubiquitination by in vitro ubiquitination assays. For this purpose, we used recombinant UBE2K in combination with either RNF2 or RNF138, the main interacting E3 enzymes of

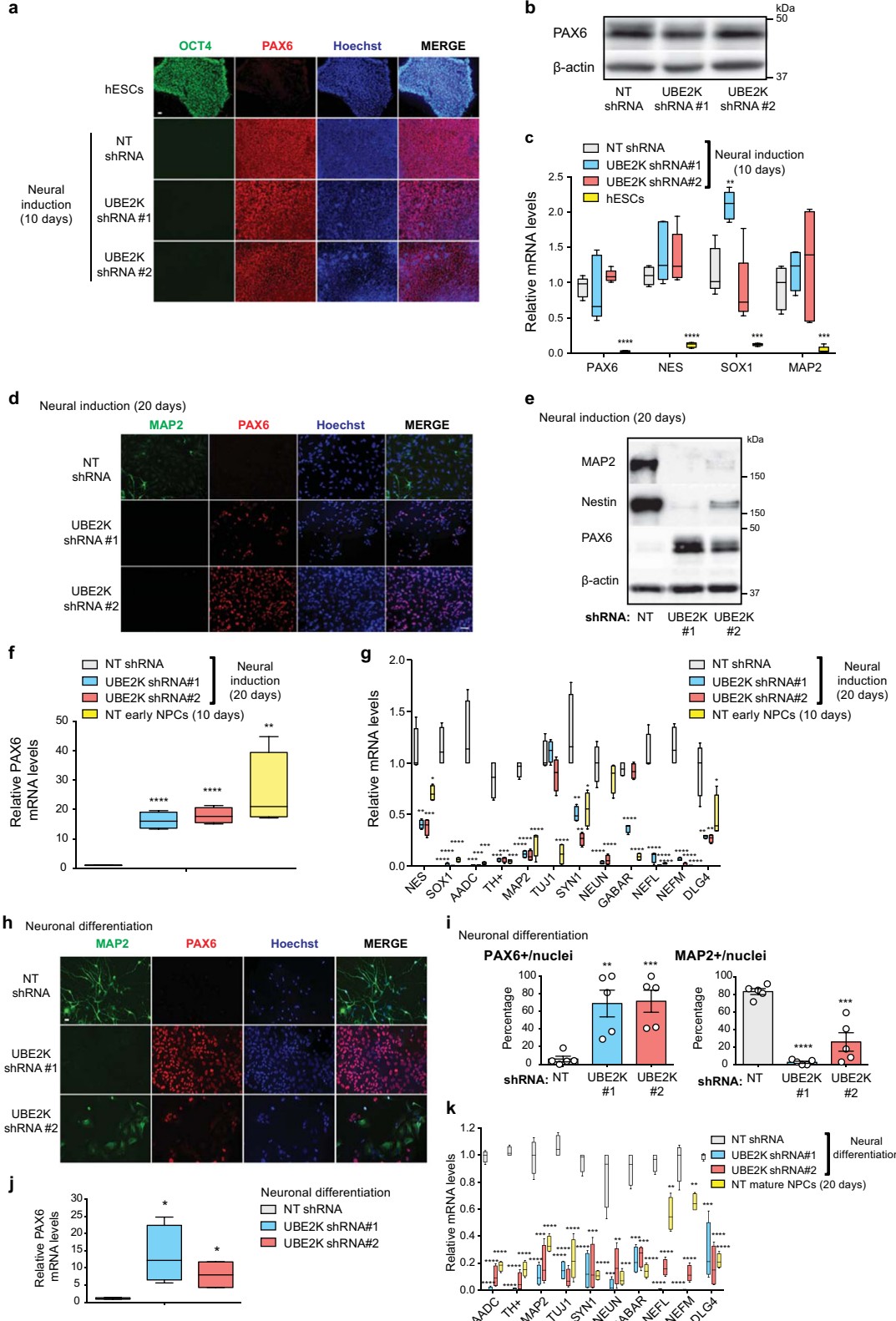

UBE2K[32–36]. The combination of UBE2K with RNF2 did not efficiently promote ubiquitination of H3. Importantly, UBE2K-RNF138 induced the robust formation of ubiquitinated H3 with a molecular weight of ~24 kDa, which could be caused by mono-ubiquitination. To a lesser extent, we also observed ubiquitinated H3 at higher molecular weights (e.g. ~48 kDa) that could correspond to polyubiquitination induced by UBE2K-RNF138

(Fig. 9a). Likewise, UBE2K-RNF138 induced ubiquitination of p53 (Fig. 9b), a previously known target of UBE2K[37]. On the contrary, we did not detect ubiquitination of histone H1 under these conditions (Fig. 9c). Thus, in vitro experiments indicate that UBE2K can promote ubiquitination of H3.

One step further was to assess whether UBE2K modulates histone H3 levels through the ubiquitin-proteasome system.

**Fig. 3 Loss of UBE2K impairs neurogenesis from hESCs. a** After 10 days of neural induction (H9 line), early NPCs were assessed by immunofluorescence with OCT4 and PAX6 staining. Hoechst staining was used as a marker of nuclei. Scale bar represents 20 μm. Images are representative of three independent experiments. **b** Western blot analysis of early H9 NPCs (10 days on neural induction) with antibody to PAX6. β-actin is the loading control. The images are representative of four independent experiments. **c** qPCR after 10 days of neural induction. Graph (relative expression to NT shRNA H9 cells) represents the mean ± s.e.m. of five independent experiments. **d** After 20 days of neural induction (H9 line), cells were assessed by immunofluorescence with MAP2 and PAX6 staining. Scale bar represents 40 μm. The images are representative of three independent experiments. **e** Western blot analysis of mature H9 NPCs (20 days on neural induction) with antibodies to MAP2, NES, PAX6 and β-actin. The images are representative of three independent experiments. **f**, **g** qPCR after 20 days of neural induction. Graphs (relative expression to NT shRNA H9 cells) represent the mean ± s.e.m. of four independent experiments. **h** After pan-neuronal differentiation treatment (H9 line), cells were assessed by immunofluorescence with MAP2 and PAX6 staining. Scale bar represents 20 μm. Images are representative of two independent experiments. **i** Percentage of PAX6 and MAP2-positive cells/total nuclei after neuronal differentiation (mean ± s.e.m. of 5 culture areas per condition of two independent experiments, NT shRNA = 349 total nuclei, UBE2K shRNA #1 = 996 total nuclei, UBE2K shRNA #2 = 370 total nuclei). **j**, **k** qPCR analysis after neuronal differentiation treatment. Graph (relative expression to NT shRNA H9 cells) represents the mean ± s.e.m. of four independent experiments. All the statistical comparisons were made by two-tailed Student's *t* test for unpaired samples. P value: *P < 0.05, **P < 0.01, ***P < 0.001, ****P < 0.0001.

Whereas ectopic expression of UBE2K did not affect global proteasome activity (Supplementary Fig. 13), it reduced total H3 levels and H3K9me3 in human HEK293 cells (Fig. 9d). Notably, the treatment with the proteasome inhibitor MG-132 diminished the effects on histone H3 induced by UBE2K overexpression (Fig. 9d and Supplementary Fig. 13), indicating that UBE2K determines proteasomal degradation of H3. Active proteasomes exist in distinct forms, but its major assembly is formed through the interaction of the catalytic 20S core with the 19S regulatory particle, generating 26S proteasome complexes that degrade polyubiquitinated proteins[15]. The 19S subunit PSMD11 is a crucial regulator of 26S proteasome assembly and activity[21,22,38]. Besides 19S, the 20S catalytic core can also be activated by other regulatory particles such as PA200 (PSME4) or PA28γ (PSME3), which promote ubiquitin-independent proteolytic degradation[15]. Since proteasome inhibitors such as MG-132 target the catalytic subunits of the 20S particle, they can inhibit all types of proteasomes. To determine which is the primary type of proteasome involved in histone H3 degradation, we knocked down specific regulators of distinct assemblies in HEK293 cells (i.e., PSMD11, PSME3 and PSME3). Since PSMD11 is essential for cell viability[22,38], we induced a mild knockdown (40%) to circumvent potential indirect effects. By these means, we generated stable PSMD11 shRNA cells that could replicate continuously and did not show obvious phenotypes in cell morphology, viability and proliferation when compared with control non-targeting shRNA stable cells. Importantly, this mild PSMD11 knockdown was sufficient to induce a slight (approx. 24%) but statistically significant decrease in proteasome activity (Fig. 9e, f). Likewise, knockdown of PSME4/PA200 induced a similar downregulation in proteasome activity (Fig. 9e, f). However, a potent knockdown of PSME3 (>90%) did not impair proteasome activity, indicating that PA28γ/20S assemblies do not contribute to proteasome activity in these cells (Fig. 9e, f). Notably, knockdown of PSMD11 induced a substantial increase in H3 levels (Fig. 9g). On the contrary, loss of either PSME4 or PSME3 did not cause significant changes in total histone H3 levels (Fig. 9g). Thus, these data indicate a prominent role of 26S proteasomes in the proteolysis of histone H3. Moreover, PSMD11 knockdown diminished the degradation of H3 induced by UBE2K overexpression (Fig. 9h). Since UBE2K can form chains of at least four Lys-48-linked ubiquitin moieties that trigger protein degradation by the 26S proteasome[39,40], we performed immunoprecipitation of H3 and examined its ubiquitination status upon UBE2K overexpression. Prior to immunoprecipitation, we treated the cells with proteasome inhibitor to block the degradation of H3 induced by UBE2K. Under these conditions, we immunoprecipitated similar amounts of histone H3 in both control and UBE2K-overexpressing cells, but UBE2K overexpression slightly increased H3 polyubiquitination (Supplementary Fig. 14a, b). To further exclude contaminating proteins following the first immunoprecipitation, we performed a re-immunoprecipitation with anti-H3 antibody. This assay supported polyubiquitination of histone H3 induced by UBE2K (Fig. 9i). By using an antibody against ubiquitin to detect all the forms of ubiquitinated H3, we could confirm polyubiquitination of H3 (Supplementary Fig. 15). However, we could not detect monoubiquitinated H3 in control or UBE2K overexpression conditions (Supplementary Fig. 15), suggesting that UBE2K mainly modulates H3 polyubiquitination in human cells. Altogether, our results indicate that UBE2K regulates H3 degradation by the 26S proteasome.

**UBE2K controls H3 levels and H3K9me3 in *C. elegans* germline.** To examine whether UBE2K modulates histone H3 in vivo, we used the nematode *Caenorhabditis elegans* as a model organism. Notably, loss of *ubc-20*, the worm orthologue of UBE2K[41], induced an increase in both total histone H3 and H3K9me3 levels of wild-type worms (Fig. 10a). In contrast, knockdown of a distinct E2 enzyme (i.e., *ubc-22*) did not affect histone H3 levels. Adult *C. elegans* contains 959 somatic cells and a germline with proliferative germ stem cells and gametes. Similar to hESCs, germ cells also rely on increased proteostasis mechanisms such as the ubiquitin-proteasome system[23,42–44]. To assess whether the regulation of histone H3 by *ubc-20* occurs in the germ cells, we compared germline-lacking worms with control sterile worms that conserve their germline[38]. As wild-type worms, knockdown of *ubc-20* increased histone H3 in control sterile worms with germline (Fig. 10b). On the contrary, we did not observe these effects in germline-lacking worms, indicating that *ubc-20* particularly modulates histone H3 in germ cells (Fig. 10b). Besides total histone H3 amounts, *ubc-20* also increased H3K9me3 levels in the germline (Fig. 10b). To further assess the role of *ubc-20* in H3K9 trimethylation, we examined *met-2* mutant worms. *met-2* is the orthologue of SETDB1 and SETDB2, two of the main H3K9 trimethylases[45,46]. Similar to wild-type worms, knockdown of *ubc-20* resulted in upregulated histone H3 levels in *met-2* mutant worms (Fig. 10c). However, loss of *ubc-20* did not increase the H3K9me3/H3 ratio in *met-2* mutants (Fig. 10c), supporting that H3K9 trimethylation induced by loss of *ubc-20* requires an active role of trimethylases. Taken together, our results suggest that the effects of UBE2K on histone H3 could be evolutionary conserved, as the worm orthologue *ubc-20* also modulates histone H3 levels and H3K9 trimethylation in the *C. elegans* germline.

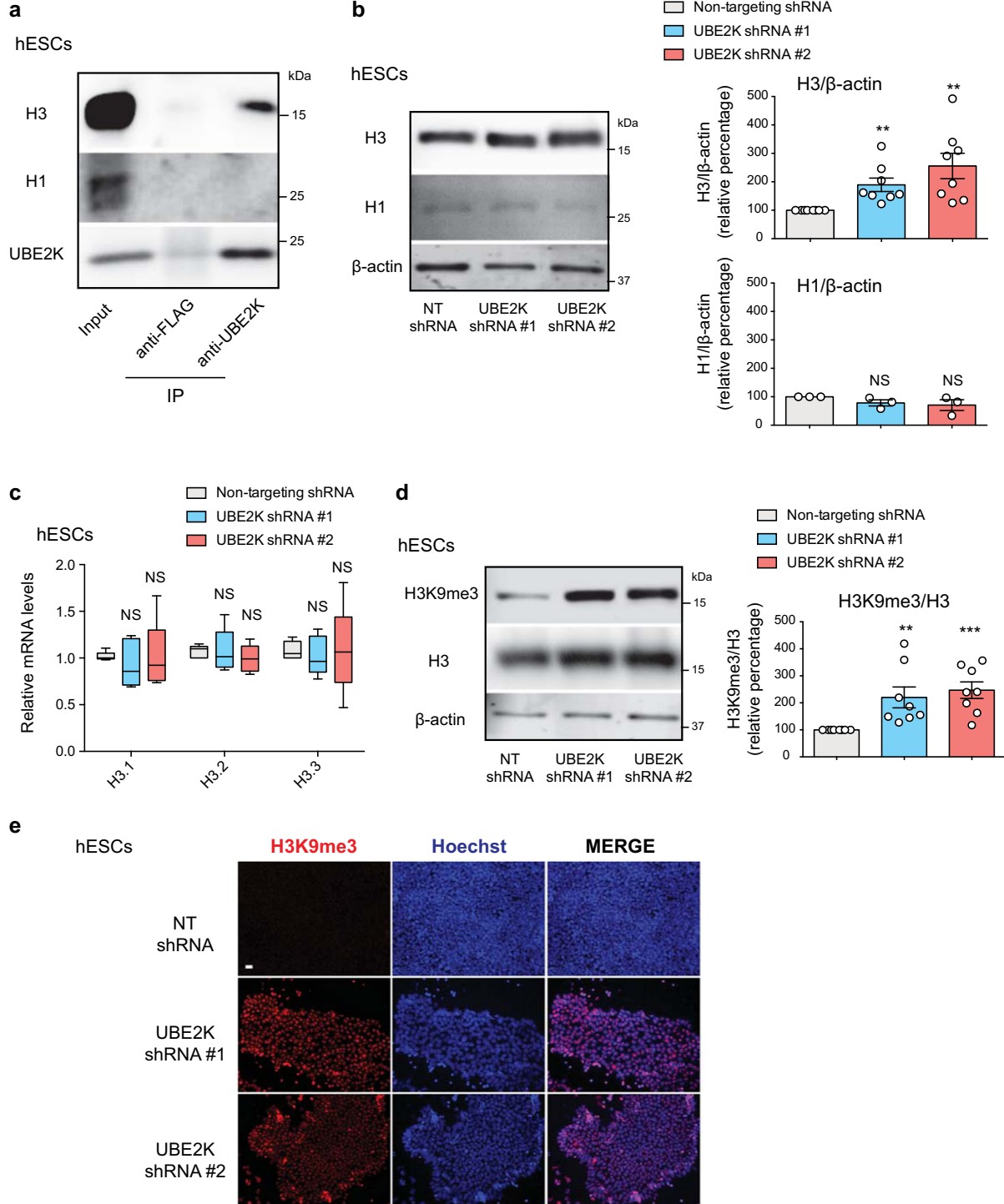

**Fig. 4 Loss of UBE2K increases total H3 levels and H3K9 trimethylation in hESCs. a** Immunoprecipitation with UBE2K and control FLAG antibodies in H9 hESCs followed by western blot with H3, H1 and UBE2K antibodies. The images are representative of three independent experiments. **b** Western blot analysis of H9 hESCs with antibodies to total histone H3 and H1. Relative percentage values of H3/β-actin (mean ± s.e.m., eight independent experiments) and H1/β-actin (mean ± s.e.m., three independent experiments) to NT shRNA control hESCs are presented. **c** qPCR analysis of histone H3 variants in H9 hESCs. Graph (relative expression to NT shRNA control hESCs) represents the mean ± s.e.m. of five independent experiments. **d** Western blot analysis of H9 hESCs with antibodies to H3K9me3 and total H3. Graph represents the relative percentage of H3K9me3/H3 ratio to NT shRNA control hESCs (mean ± s.e.m., eight independent experiments). **e** Immunocytochemistry of H9 hESCs with antibody to H3K9me3. Hoechst staining was used as a marker of nuclei. Scale bar represents 20 μm. The images are representative of three independent experiments. All the statistical comparisons were made by two-tailed Student's *t* test for unpaired samples. *P* value: **\*\*P* < 0.01, \*\*\*P* < 0.001. NS not significant (*P* > 0.05).

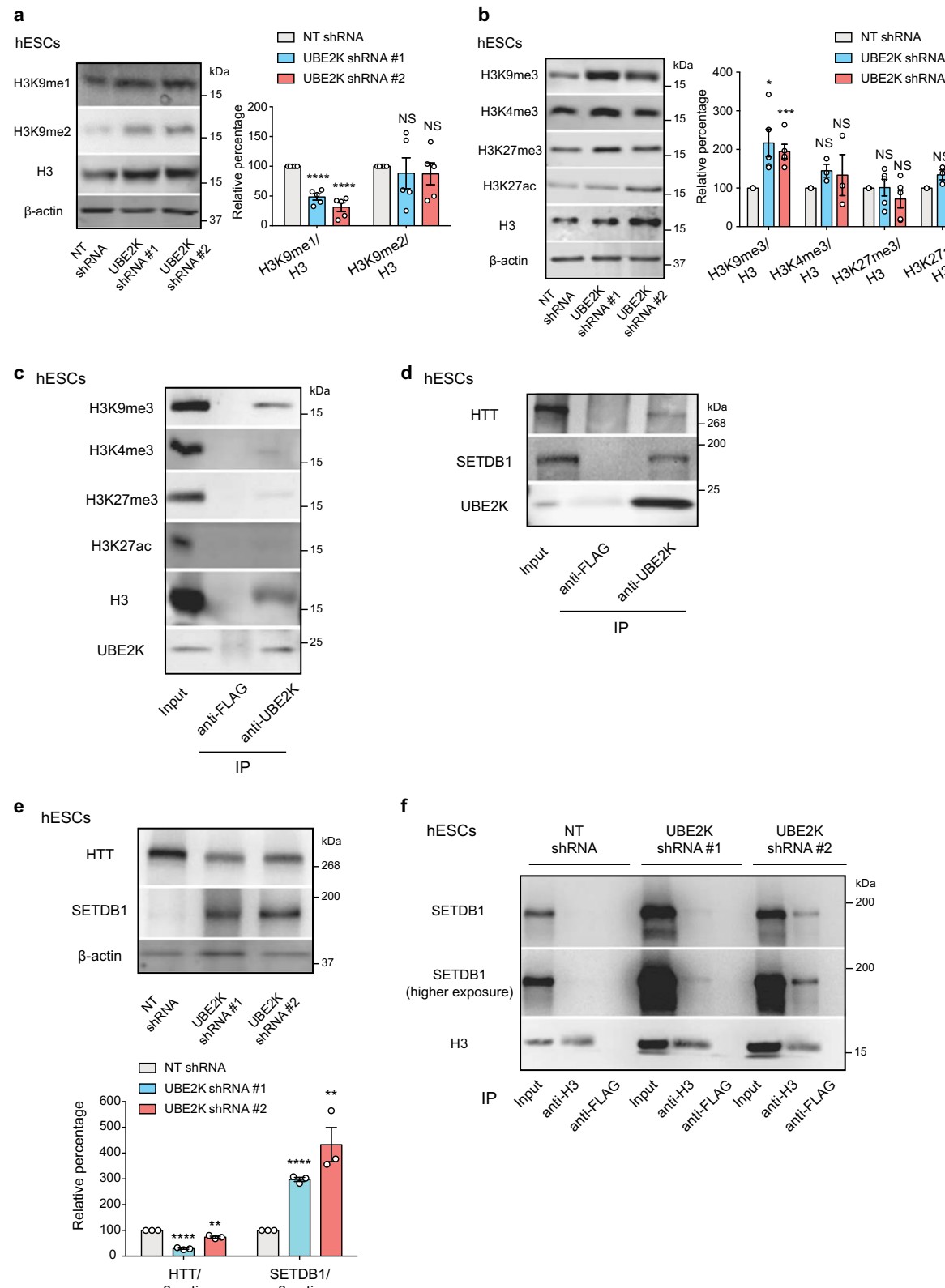

## Discussion

With the essential role of histones in chromatin compaction and gene expression, it is of central interest to define not only the regulatory mechanisms of histone modification, but also total histone amounts. During spermatogenesis, histones are replaced by transition proteins and protamines in post-meiotic cells, a process mediated by the ubiquitin-proteasome system[47].

Moreover, somatic cells also exhibit replacement of histones at promoter regions or active gene bodies[48]. Importantly, an excess of histones impairs transcription, increases DNA-damage sensitivity, and promotes either chromosome aggregation or loss[49]. In yeast, excess histones are rapidly degraded by the proteasome in a pathway involving the E2 enzyme Ubc4/5 and E3 ligases such as Tom1 and Hel1[49,50].

**Fig. 5 Knockdown of UBE2K increases SETDB1 levels in hESCs. a** Western blot analysis of H9 hESCs with antibodies to H3K9me1, H3K9me2, total H3 and β-actin. Graph represents the H3K9me1/H3 and H3K9me2/H3 relative percentage values to NT shRNA hESCs (mean ± s.e.m., five independent experiments). **b** Western blot analysis of H9 hESCs with antibodies to H3K9me3, H3K4me3, H3K27me3, H3K27ac, total H3 and β-actin. Graph represents the mean ± s.e.m of H3K9me3/H3 ($n = 5$ independent experiments), H3K4me3/H3 ($n = 3$), H3K27me3/H3 ($n = 5$) and H3K27ac/H3 ($n = 3$) relative percentage values to NT shRNA hESCs. **c** Immunoprecipitation with UBE2K and control FLAG antibodies in H9 hESCs followed by western blot with H3K9me3, H3K4me3, H3K27me3, H3K27ac, H3 and UBE2K antibodies. The images are representative of three independent experiments. **d** Immunoprecipitation with UBE2K and control FLAG antibodies in H9 hESCs followed by western blot with HTT, SETDB1 and UBE2K antibodies. The images are representative of three independent experiments. **e** Western blot analysis of H9 hESCs with antibodies to HTT and SETDB1. The graph represents the relative percentage values (corrected for β-actin loading control) to NT shRNA control hESCs (mean ± s.e.m., three independent experiments). **f** Immunoprecipitation with H3 and control FLAG antibodies in H9 hESCs followed by western blot with H3 and SETDB1 antibodies. The images are representative of two independent experiments. All the statistical comparisons were made by two-tailed Student's $t$ test for unpaired samples. $P$ value: *$P < 0.05$, **$P < 0.01$, ***$P < 0.001$, ****$P < 0.0001$. NS not significant ($P > 0.05$).

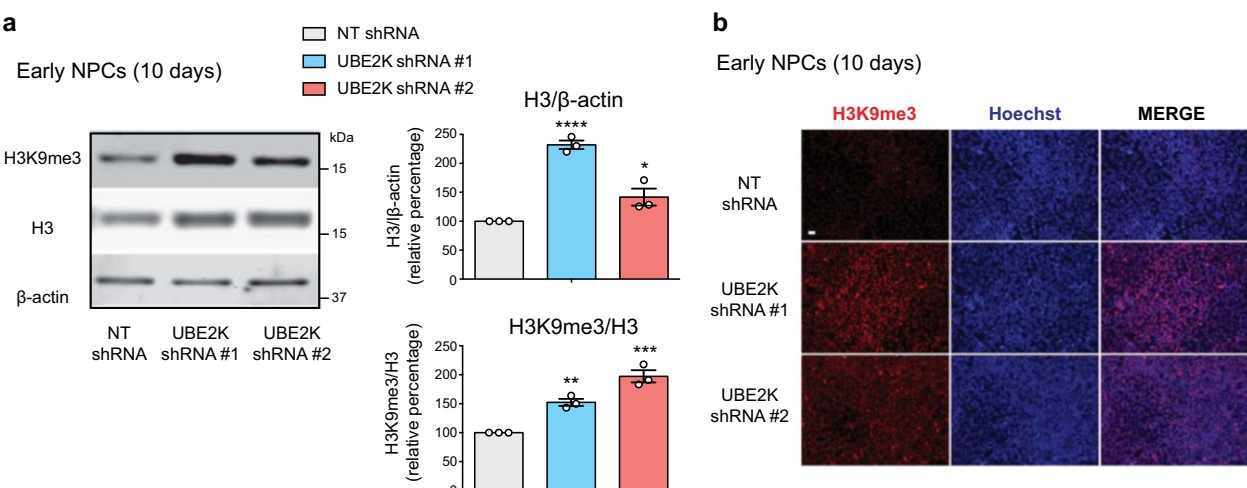

**Fig. 6 NPCs derived from UBE2K shRNA hESCs retain alterations in both total H3 levels and H3K9me3/H3 ratio. a** After 10 days of neural induction treatment, early H9 NPCs were analysed by western blot with antibodies to total H3 and H3K9me3. Graphs represent the histone H3/β-actin and H3K9me3/H3 relative percentage values to NT shRNA NPCs (mean ± s.e.m., three independent experiments). **b** After 10 days of neural induction, early H9 NPCs were assessed by immunofluorescence with H3K9me3 and Hoechst staining. Scale bar represents 20 μm. All the statistical comparisons were made by two-tailed Student's $t$ test for unpaired samples. $P$ value: *$P < 0.05$, **$P < 0.01$, ***$P < 0.001$, ****$P < 0.0001$.

Despite their essential biological role, little is known about how histone levels are regulated in multicellular organisms. As the origin of multicellular organisms, pluripotent stem cells have stringent mechanisms to protect their genome[51] and proteome[18]. Since hESCs exhibit an endogenous epigenetic landscape with a more open and dynamic architecture of chromatin[52,53], we hypothesized that these cells can also have intrinsic mechanisms to maintain the homeostasis of histone levels. For this purpose, we examined whether their enhanced ubiquitin-proteasome system is involved in histone regulation. Our approach led us to identify high levels of UBE2K as a determinant of histone H3 protein amounts, potentially via regulating its ubiquitination and proteasomal-mediated degradation. First, we confirmed UBE2K-mediated ubiquitination of H3 by in-vitro ubiquitination assays. In these experiments, UBE2K in combination with the E3 ligase RNF138 robustly induces the formation of ubiquitinated H3 with a molecular weight of ~24 kDa, which could indicate attachment of one ubiquitin. To a lesser extent, UBE2K-RNF138 also triggers the formation of ubiquitinated H3 with higher molecular weights. Among them, we found a ubiquitinated H3 form of ~48 kDa. According to the molecular weight, this band could correspond to H3 tagged with a chain of four ubiquitins, which is a mark for proteasomal recognition and degradation. However, in-vitro ubiquitination experiments alone cannot discard that the distinct ubiquitinated H3 forms detected in the assay might be caused by multi-monoubiquitination. Remarkably, immunoprecipitation of histone H3 supports that UBE2K overexpression

triggers polyubiquitination of H3 in human cells, whereas we could not detect monoubiquitinated H3 in control or UBE2K overexpression conditions. In line with these findings, UBE2K overexpression results in degradation of histone H3, whereas the treatment with MG-132 proteasome inhibitor antagonizes these effects. Most importantly, a mild decrease in proteasome activity upon moderate knockdown of PSMD11, a key regulator of 26S proteasomes that degrade polyubiquitinated proteins, induces the strongest increase in total H3 levels when compared with activators of other proteasome assemblies involved in ubiquitin-independent proteolytic degradation[15]. For instance, even when PSMD11 and PSME4 knockdown induced a similar inhibition of total proteasome activity (~24%), only PSMD11 shRNA resulted in robust changes in histone H3 levels. Whereas these results support a prominent role of 26S proteasomes in H3 degradation, we cannot discard that other proteasome assemblies could also regulate H3 levels. For instance, it is possible that a stronger knockdown of PSME4 may induce a further reduction in proteasome activity, and eventually lead to changes in histone H3 levels. As a further evidence of a role of 26S proteasomes in H3 degradation, we found that PSMD11 knockdown blocks the degradation of H3 induced by UBE2K overexpression.

Besides total H3 levels, UBE2K also modulates H3K9 tri-methylation, a histone modification associated with hetero-chromatin formation and gene repression. Our data indicate that UBE2K impinges upon H3K9 trimethylation by two mechanisms

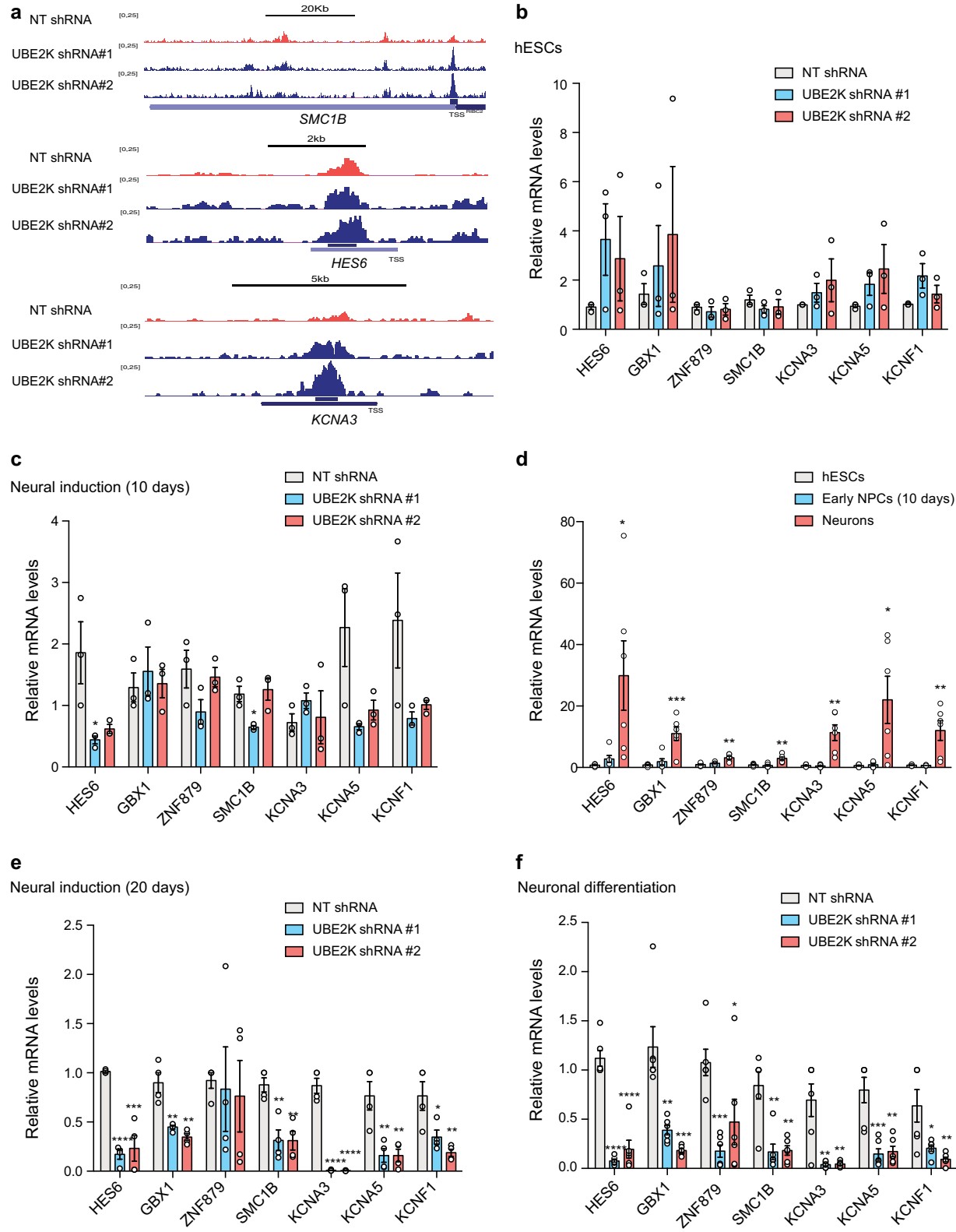

converging on SETDB1. First, we find that loss of UBE2K decreases the levels of HTT, an inhibitor of SETBD1 activity. In these lines, our results in *C. elegans* support that H3K9 tri-methylation upon UBE2K knockdown requires the activity of trimethylases such as SETDB1. Besides changes in HTT, knock-down of UBE2K results in higher SETDB1 protein levels. Given the important role of HTT and SETDB1 in gene expression, it will

be fascinating to define how UBE2K regulates their levels. Since UBE2K interacts with SETDB1, an intriguing hypothesis is that UBE2K directly modulates its ubiquitination and proteasomal degradation. Another possibility is that UBE2K regulates SETDB1 levels via its recruitment into histone complexes, preventing its degradation. To address these hypothesis, it will also be important to assess whether UBE2K-mediated ubiquitination and stability of

**Fig. 7 Altered H3K9me3 marks in hESCs upon UBE2K knockdown impairs the induction of distinct neural genes. a** H3K9me3 ChIP-seq profiles of *SMC1B, HES6* and *KCNA3* generated in NT and UBE2K shRNA H9 hESCs. **b** qPCR analysis of H9 hESCs. Graph (relative expression to NT shRNA) represents the mean ± s.e.m. of three independent experiments. No significant differences were found. **c** qPCR analysis after 10 days of neural induction of H9 hESCs. Graph (relative expression to NT shRNA) represents the mean ± s.e.m. of three independent experiments. **d** qPCR analysis of wild-type H9 hESCs and their differentiated counterparts. Graph (relative expression to NT shRNA) represents the mean ± s.e.m. of six independent experiments. **e** qPCR analysis after 20 days of neural induction of H9 hESCs. Graph (relative expression to NT shRNA) represents the mean ± s.e.m. of four independent experiments. **f** qPCR analysis after neuronal differentiation (H9 line). Graph (relative expression to NT shRNA) represents the mean ± s.e.m. of six independent experiments. Statistical comparisons were made by two-tailed Student's *t* test for unpaired samples. *P* value: *\*P* < 0.05, *\*\*P* < 0.01, *\*\*\*P* < 0.001, *\*\*\*\*P* < 0.0001.

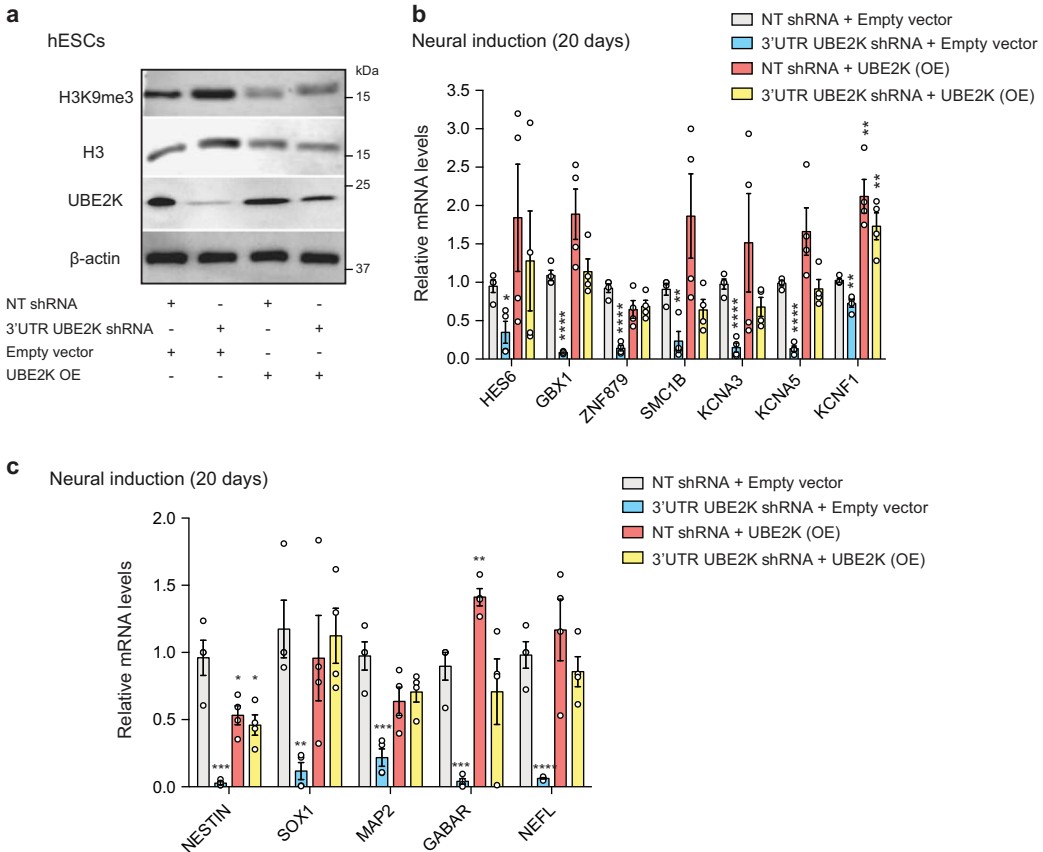

**Fig. 8 Ectopic expression of UBE2K in UBE2K-deficient hESCs rescues H3K9me3 levels and induction of neuronal genes. a** Western blot analysis of H9 hESCs with antibodies to H3K9me3, total H3, UBE2K and β-actin. The images are representative of two independent experiments. **b, c** qPCR analysis after 20 days of neural induction of H9 hESCs. Graph (relative expression to NT shRNA) represents the mean ± s.e.m. of four independent experiments. All the statistical comparisons were made by two-tailed Student's *t* test for unpaired samples. *P* value: *\*P* < 0.05, *\*\*P* < 0.01, *\*\*\*P* < 0.001, *\*\*\*\*P* < 0.0001.

H3 determines changes in the total protein amounts or interactions of key regulators of H3K9 trimethylation.

Since UBE2K particularly increased the H3K9me3 ratio, we focused on the impact of upregulated H3K9me3 in neurogenesis from hESCs. Importantly, loss of UBE2K not only increases global H3K9 trimethylation but also H3K9me3 enrichment in genes required for neural and neuronal differentiation such as GBX1, HES6 or distinct potassium voltage-gated channels. It is important to note that these changes do not have strong effects on hESCs and their commitment into early NPCs. However, H3K9me3 enrichment in the aforementioned genes diminishes their induction during differentiation into mature NPCs and neurons. Interestingly, a recent study found that low levels of H3K9me3 in hESCs allow for the establishment of compacted heterochromatin in specific protein-coding-genes at later stages, a critical process for cell differentiation[8]. Thus, UBE2K could contribute to maintaining low H3K9me3 levels in hESCs, defining

their differentiation potential. In these lines, we observed that loss of UBE2K in hESCs does not affect the first steps of differentiation into early NPCs, but impairs their ability to differentiate into mature NPCs and neurons. We speculate that the H3K9me3-mediated repression of genes such as GBX1, HES6, KCNA3, KCNA5 could contribute to these phenotypes, including the lack of induction in distinct neural and neuronal markers during mid and late stages of differentiation. For instance, UBE2K knockdown does not affect the expression of the neural markers Nestin and SOX1 in early NPCs, but blocks their further induction during differentiation into mature NPCs. When compared with early NPCs, control mature NPCs also have increased levels of all the neuronal markers tested (i.e., AADC, TH+, MAP2, TUJ1, SYN1, NEUN, GABAR, NEFL, NEFM, DLG4). Importantly, loss of UBE2K in hESCs is sufficient to inhibit the induction of these factors during differentiation into mature NPCs, with the exception of TUJ1. Moreover, UBE2K knockdown also blocked

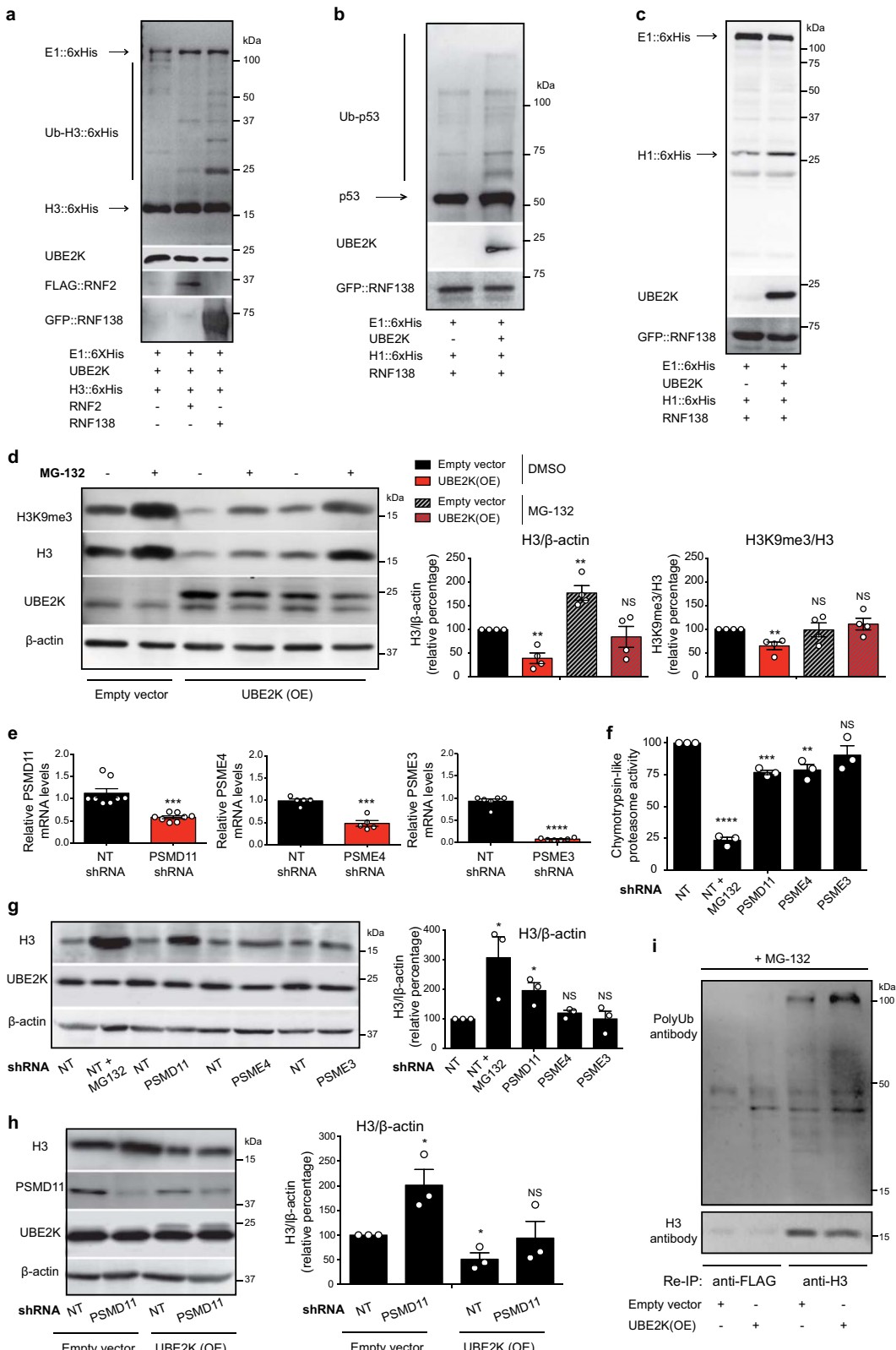

the further induction of all the tested neuronal markers during terminal differentiation into neurons, including TUJ1. Despite the delay in TUJ1 alterations, our results indicate that UBE2K knockdown already has deleterious effects during differentiation into mature NPCs. Although our data suggest an important role of H3K9me3 changes in neuronal differentiation, it is uncertain whether H3K9 trimethylation or H3 stability is the key player in

regulating the neurogenic potential of hESCs. For instance, dysregulated H3 levels may also contribute to diminishing neurogenic potential and even has a more prominent role in this phenotype relative to H3K9me3 changes. Whereas UBE2K knockdown does not significantly change the ratio of H3K4me3, H3K27me3, H3K27ac when normalized to total H3, it increases overall levels of these modifications. Thus, we cannot discard that

**Fig. 9 UBE2K modulates ubiquitination and proteasomal degradation of histone H3. a** In vitro ubiquitination assay of 6xHis-tagged H3F3A with UBE2K and FLAG::RNF2 or GFP::RNF138 ubiquitin ligases followed by immunoblotting with antibodies to 6xHis, UBE2K, FLAG and GFP. The images are representative of two independent experiments. **b** In vitro ubiquitination of recombinant p53 followed by immunoblotting with antibodies to p53, UBE2K, and GFP. The images are representative of two independent experiments. **c** In vitro ubiquitination of 6xHis:H1 followed by immunoblotting with antibodies to 6xHis, UBE2K, and GFP. The images are representative of two independent experiments. **d** Western blot of UBE2K overexpressing (OE) HEK293 with antibodies to H3, H3K9me3 and β-actin. The graphs represent the relative percentage of H3/β-actin and H3K9me3/H3 to DMSO-empty vector cells (mean ± s.e.m. of four independent experiments). When indicated in the figure, cells were treated with 5 μM MG-132 for 16 h. **e** Knockdown levels of proteasome activators in HEK293 cells. The graph (relative expression to non-targeting (NT) shRNA HEK293 cells) represents the mean ± s.e.m. (PSMD11 shRNA ($n = 8$), PSME4 shRNA ($n = 5$), PSME3 shRNA ($n = 6$)). **f** Percentage of chymotrypsin-like proteasome activity relative to NT shRNA HEK293 cells (mean ± s.e.m. of three independent experiments). MG-132 treatment: 5 μM MG-132 for 16 h. **g** Western blot of HEK293 cells with antibodies to H3 and UBE2K. The graph represents the relative percentage of H3/β-actin to NT shRNA cells (mean ± s.e.m. of three independent experiments). **h** Western blot of HEK293 cells with antibodies to H3, PSMD11 and UBE2K. The graph represents the relative percentage of H3/β-actin to NT shRNA + empty vector cells (mean ± s.e.m. of three independent experiments). **i** After immunoprecipitation with anti-H3 and anti-FLAG antibodies in HEK293 cells, we performed a re-immunoprecipitation (Re-IP) with the same antibodies. Re-IP was followed by western blot with antibodies against H3 and polyubiquitinated proteins (polyUb) to detect immunoprecipitated H3 protein and polyUb-H3, respectively. The images are representative of two independent experiments. Prior to immunoprecipitation, cells were treated with 5 μM MG-132 (16 h). All the statistical comparisons were made by two-tailed Student's $t$-test for unpaired samples. $P$ value: $*P < 0.05$, $**P < 0.01$, $***P < 0.001$, $****P < 0.0001$. NS not significant.

upregulated H3K4me3, H3K27me3, H3K27ac amounts resulting from increased H3 levels also have a key role in regulating neurogenic potential. Another intriguing possibility is that UBE2K-mediated changes in H3 stability are the upstream signal to modulate H3K9 trimethylation. In this speculative model, changes in H3 levels could be considered as the key determinant of neurogenesis, because they would precede H3K9 trimethylation.

Remarkably, our experiments in adult *C. elegans* indicate that UBE2K knockdown particularly affects histone H3 and H3K9 trimethylation in the immortal germline, which generates gametes for reproduction. On the other hand, we observed that enhancing the levels of UBE2K in human cell lines is sufficient to mimic histone H3 modulation of pluripotent cells. Thus, our results indicate that a precise regulation of UBE2K levels could contribute to determining cell-type epigenetic landscapes.

## Methods

**hESC/iPSC culture and differentiation.** Both H9 (WA09) and H1 (WA01) hESC lines were obtained from the WiCell Research Institute. The human iPSC line (ACS-1011) and their parental HFF-1 fibroblasts (SCRC-1041) were obtained from ATCC. hESCs/iPSCs were maintained on Geltrex-coated plates (ThermoFisher Scientific) using mTeSR1 media (Stem Cell Technologies). Undifferentiated hESC/iPSC colonies were passaged using a solution of dispase (2 mg ml⁻¹), and scraping the colonies with a glass pipette. All the cell lines were tested for mycoplasma contamination at least once every three weeks and no mycoplasma contamination was detected. Research involving hESCs was performed with approval of the German Federal competent authority (Robert Koch Institute). The H9 and H1 hESCs used in this study were authenticated by short tandem repeat (STR) profile across 8 STR loci[17].

Neural differentiation was performed following the monolayer culture method with STEMdiff Neural Induction Medium (Stem Cell Technologies) based on ref. [54]. Undifferentiated hESCs were rinsed once with PBS and then dissociated with Gentle Dissociation Reagent (Stem Cell Technologies) for 10 min. Then, we gently dislodged the cells and added 2 ml of Dulbecco's Modified Eagle Medium (DMEM)-F12 (ThermoFisher Scientific) + 10 μM ROCK inhibitor (Abcam). Cells were then centrifuged at 300×g for 10 min and resuspended on STEMdiff Neural Induction Medium supplemented with 10 μM ROCK inhibitor. Cells were plated on polyornithine (15 μg ml⁻¹)/laminin (10 μg ml⁻¹)-coated plates at a density of 200,000 cells cm⁻². As indicated in the corresponding figures, hESCs were cultured on neural induction medium for a total of 10 and 20 days to generate early NPCs (PAX6-positive cells) and mature NPCs with the ability to efficiently differentiate into neurons, respectively.

For neuronal differentiation, mature NPCs (20 days) were dissociated with Accutase (Stem Cell Technologies) and transferred to polyornithine/laminin-coated plates with neuronal differentiation medium Dulbecco's Modified Eagle Medium (DMEM)/F12, B27, N2 (ThermoFisher Scientific), 1 μg ml⁻¹ laminin (ThermoFisher Scientific), 20 ng ml⁻¹ BDNF (Peprotech), 20 ng ml⁻¹ GDNF (Peprotech), 1 mM dibutyryl-cyclic AMP (Sigma) and 200 nM ascorbic acid (Sigma)[22]. Cells were differentiated for 1 month, with weekly feeding of fresh neuronal differentiation medium. Endoderm differentiation of H9 hESCs was performed using STEMdiff Definitive Endoderm Kit (Stem Cell Technologies). For mesodermal differentiation, hESC colonies were dissociated with Accutase and

single cells were seeded at a density of 50,000 cells cm⁻² in mTeSR1 media (Stem Cell Technologies) supplemented with 10 μM ROCK inhibitor. One day after, we replaced mTeSR1 media with STEMdiff Mesoderm Induction Medium (Stem Cell Technologies). Then, cells were fed daily with STEMdiff Mesoderm Induction Medium and collected for experiments after 5 days.

**Lentiviral infection of hESCs.** Lentivirus (LV)-non-targeting shRNA control, LV-UBE2K shRNA #1 (TRCN0000237896), LV-UBE2K shRNA #2 (TRCN0000237893), LV-3′-UTR UBE2K shRNA (TRCN0000237895), LV-PSMD11 shRNA (TRCN0000003950), LV-PSME4 shRNA (TRCN0000158223) and LV-PSME3 shRNA (TRCN0000290094) in pLKO.1-puro vector were obtained from Mission shRNA (Sigma). To establish stable shRNA hESC lines, hESCs growing on Geltrex-coated plates were incubated with mTesR1 medium supplemented with 10 μM ROCK inhibitor for 1 h and individualized with Accutase. 50 000 cells were infected with 20 μl of concentrated lentivirus in the presence of 10 μM ROCK inhibitor for 1 h. Cell suspension was centrifuged to remove virus, passed through a mesh of 40 μM to obtain individual cells, and plated on a feeder layer of mitotically inactive mouse embryonic fibroblasts (MEFs) in hESC media (DMEM/F12, 20% knockout serum replacement (ThermoFisher Scientific), 0.1 mM non-essential amino acids, 1 mM L-glutamine, 10 ng ml⁻¹ bFGF (Joint Protein Central) and β-mercaptoethanol) supplemented with 10 μM ROCK inhibitor. When small hESC colonies arose after a few days in culture, we performed 1 μg ml⁻¹ puromycin selection during 2 days and colonies were manually passaged onto MEFs to generate stable knockdown hESC lines. Once stable knockdown hESC lines were established, the cells were maintained on Geltrex-coated plates using mTeSR1 media. To establish stable shRNA HEK293 lines, HEK293 cells (ATCC, #CRL-11268) were transduced with 5 μl of concentrated lentivirus and selected by adding puromycin at a concentration of 2 μg ml⁻¹.

For the generation of UBE2K-overexpressing lentiviral constructs (UBE2K (OE)), human UBE2K complementary DNA was PCR-amplified and cloned into CD522A-1 pCDH cDNA Cloning Lentivector (System Biosciences) using NheI and BamHI. This construct was sequence verified and thereafter transfected into packaging cells to produce high titer lentiviruses. As a control, we generated lentiviral particles of CD522A-1 pCDH lacking UBE2K insert (empty vector). HEK293 cells were transduced with either empty vector or UBE2K(OE) lentiviruses and selected by adding puromycin at a concentration of 2 μg ml⁻¹. To overexpress UBE2K in stable cell lines that were previously infected with lentivirus expressing shRNA and selected for puromycin resistance, we cloned human UBE2K into CD515B-1 pCDH cDNA Cloning Lentivector (System Biosciences), a vector that contains hygromycin resistance gene instead of puromycin resistance gene. As a control for these experiments, we generated lentiviral particles of CD515B-1 pCDH lacking UBE2K insert (empty vector). Cells were selected by adding hygromycin at a concentration of 100 μg ml⁻¹.

**RNA sequencing.** We extracted RNA using RNAbee (Tel-Test Inc.). Libraries were then prepared using the TruSeq Stranded mRNA Library Prep Kit. Library preparation started with 1 μg total RNA. After selection with poly-T oligo-attached magnetic beads, mRNA was purified and fragmented using divalent cations under elevated temperature. The RNA fragments underwent reverse transcription with random primers followed by second strand cDNA synthesis with DNA Polymerase I and RNase H. After end repair and A-tailing, indexing adapters were ligated. Then, products were purified and amplified (20 μl template, 14 PCR cycles) to generate the final cDNA libraries. Following library validation and quantification (Agilent 2100 Bioanalyzer), we pooled equimolar amounts of library. Then, the pool was quantified by the Peqlab KAPA Library Quantification Kit and Applied

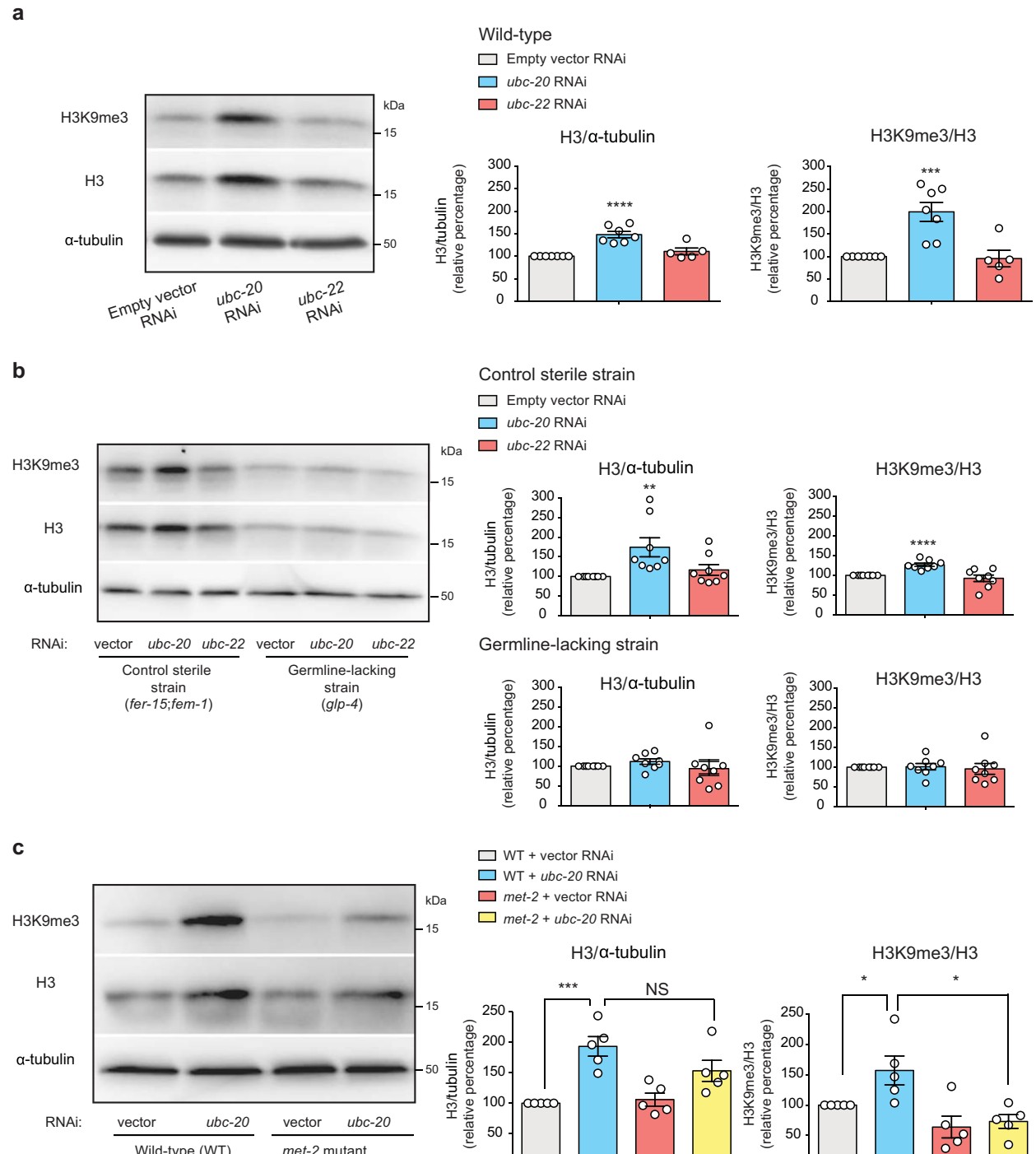

**Fig. 10 *ubc-20*, a *C. elegans* orthologue of UBE2K, regulates histone H3 and H3K9me3 levels in the germline. a** Western blot analysis of wild-type *C. elegans* upon knockdown of either the UBE2K orthologue (*ubc-20*) or a distinct E2 enzyme (*ubc-22*) with antibodies to total H3, H3K9me3 and α-tubulin loading control. Graphs represent the relative percentage values of H3/α-tubulin and H3K9me3/H3 to empty vector RNAi-treated worms (mean ± s.e.m., empty vector RNAi *n* = 7 independent experiments, *ubc-20* RNAi *n* = 7, *ubc-22* RNAi *n* = 5). RNAi was started at day 1 of adulthood. **b** Western blot analysis of control sterile worms (*fer-15(b26);fem-1(hc17)*) and germline-lacking worms (*glp-4(bn2)*) with antibodies to total H3, H3K9me3 and α-tubulin loading control. The graphs represent the relative percentage values of H3/α-tubulin and H3K9me3/H3 to the corresponding empty vector RNAi-treated strain (mean ± s.e.m. of 8 independent experiments). **c** Western blot analysis of wild-type and *met-2(n4256)* mutant worms upon knockdown of *ubc-20*. The graphs represent the relative percentage values of H3/α-tubulin and H3K9me3/H3 to empty vector RNAi-wild-type worms (mean ± s.e.m. of 5 independent experiments). All the statistical comparisons were made by two-tailed Student's *t*-test for unpaired samples. *P* value: **P* < 0.05, ***P* < 0.01, ****P* < 0.001, *****P* < 0.0001. NS not significant.

Biosystems 7900HT Sequence Detection System. The pool was sequenced on an Illumina HiSeq 4000 sequencer with a paired- end (2 × 75 bp) protocol.

RNA-sequencing data were analyzed using a QuickNGS pipeline[55]. We performed basic read quality check and read statistics by using FastQC and SAMtools, respectively. The basic data processing consists of a splicing-aware alignment using Tophat2[56] followed by reference-guided transcriptome reassembly with Cufflinks2[57,58]. Read count means, fold-change (FC) and *P* values were calculated with DEseq2[59], whereas gene expression for the individual samples was calculated with Cufflinks2[57,58] as FPKMs, using in both cases genomic annotation from the Ensembl database (version 87).

**Sample preparation for quantitative proteomics and analysis**. For the comparison between control and UBE2K shRNA H9 hESCs, label-free quantitative (LFQ) proteomics were performed as we described in ref. [12]. Cells were collected in urea buffer (8 M urea, 50 mM ammonium bicarbonate and 1× complete protease inhibitor mix with EDTA (Roche)), homogenized with a syringe and cleared using centrifugation (16,000 *g*, 20 min). Supernatants were reduced (1 mM DTT, 30 min), alkylated (5 mM iodoacetamide (IAA), 45 min) and digested with trypsin at a 1:100 w/w ratio after diluting the urea concentration to 2 M. After one day, samples were cleared (16,000 *g*, 20 min) and supernatant was acidified. Then, peptides were cleaned up using stage tip extraction[60]. The liquid chromatography tandem mass spectrometry (LC-MS/MS) equipment consisted of an EASY nLC 1000 coupled to the quadrupole based QExactive instrument (Thermo Scientific) via a nano-spray electroionization source. Peptides were separated on an in-house packed 50 cm column (1.9 µm C18 beads, Dr. Maisch) using a binary buffer system: (A) 0.1% formic acid, (B) 0.1 % formic acid in ACN. The content of buffer B was raised from 7 to 23% within 120 min and then increased to 45% within 10 min. Buffer B fraction was raised to 80% within 5 min and held for further 5 min after which it was decreased to 5% within 2 min and held there for further 3 min before the next sample was loaded onto the column. Eluting peptides were ionized by an applied voltage of 2.2 kV. The capillary temperature was 275 °C and the S-lens RF level was set to 60. MS1 spectra were acquired using a resolution of 70,000 (at 200 m/z), an Automatic Gain Control (AGC) target of 3e6 and a maximum injection time of 20 ms in a scan range of 300–1750 Th. In a data-dependent mode, the 10 most intense peaks were selected for isolation and fragmentation in the HCD cell using a normalized collision energy of 25 at an isolation window of 2.1 Th. Dynamic exclusion was enabled and set to 20 s. We used the following MS/MS scan properties: 17.500 resolution at 200 *m/z*, AGC target of 5e5 and maximum injection time of 60 ms. We analysed all LFQ proteomics data sets with MaxQuant software (release 1.5.3.8). We used the LFQ mode[61] and MaxQuant default settings for protein identification and LFQ quantification. Downstream analyses of LFQ values were performed with Perseus (v. 1.5.2.4)[62].

**Protein immunoprecipitation for interactome analysis**. Protein immunoprecipitation for interactome assays were performed as we reported in ref. [63]. Cells were lysed in modified Radioimmunoprecipitation assay (RIPA) buffer (50 mM Tris-HCl (pH 7.4), 150 mM NaCl, 1 % IgPal, 0.25% sodium deoxycholate, 1 mM EDTA, 1 mM PMSF) supplemented with protease inhibitor (Roche). Lysates were centrifuged at 10,000 *g* for 10 min at 4 °C. Then, the supernatant was collected and incubated with anti-UBE2K antibody (Cell Signaling, #8226 1:50) for 30 min and subsequently with 100 µl Protein A magnetic beads (Miltenyi) for 1 h on the overhead shaker at 4 °C. As a control, the same amount of protein was incubated with anti-FLAG antibody (SIGMA, F7425, 1:100) in parallel. After the incubation with antibodies, supernatants were subjected to magnetic column purification followed by three washes using wash buffer 1 (50 mM Tris-HCl (pH 7.4), 150 mM NaCl, 5% glycerol and 0.05% IgPal). Next, columns were washed five times with wash buffer 2 (50 mM Tris-HCl (pH 7.4), 150 mM NaCl). After the washing steps, the pellet was incubated with 2x Laemmli buffer, boiled for 5 min and centrifuged 5 min at maximum speed. The supernatant was taken and loaded onto a sodium dodecyl sulfate–polyacrylamide gel electrophoresis (SDS–PAGE) gel for western blot analysis.

**Identification of ubiquitination events in histone H3**. Histones were purified using a method that combines sulfuric acid extraction with ion exchange chromatography[64,65]. Briefly, HEK293 cells were treated with 5 µM MG-132 overnight to avoid proteasomal degradation, washed twice with ice-cold PBS and lysed by rotation in 6 ml of 0.1 M $H_2SO_4$ at 4 °C for 2 h. The lysate was centrifuged at 2200 *g* at 4 °C for 20 min. The pellet with non-soluble proteins and cell debris was discarded. The supernatant was neutralized in 1 M Tris-Hcl pH 8.0 buffer. Volume of lysate was made up to 15 ml by adding binding buffer (50 mM Tris pH 8.0, 0.5 M NaCl, 2 mM EDTA, 0.25 mM PMSF, 1 mM DTT). Sulfopropyl (SP)-Sepharose resin (S1799, Sigma) was packed into a column and pre-equilibrated with 10 volumes of binding buffer. 15 ml of lysate was passed through the column. The column was then washed with 10 volumes of binding buffer and 30 volumes of washing buffer (50 mM Tris-HCl, pH 8.0, 0.6 M NaCl, 2 mM EDTA, 0.25 mM PMSF, 1 mM DTT). Proteins were eluted with elution buffer (50 mM Tris-HCl, pH 8.0, 2 M NaCl, 2 mM EDTA, 0.25 mM PMSF 1 mM DTT) in ten fractions. Eluted proteins were precipitated overnight in 4% (vol/vol) PCA at 4 °C. The fractions were then centrifuged at 21,000 *g* at 4 °C for 45 min, and the resulting pellets were

washed twice with 4% PCA, twice with 0.2% HCl in acetone and twice with acetone. The protein pellets were finally dissolved in milliQ water and protein concentration was determined by standard BCA method.

Then, 500 µg histone proteins were denatured by diluting in 8 M Urea (in 0.1 M Tris/HCl pH 8.5) in a final volume of 200 µl. Additionally, 2 µl of 1 M TCEP (final 10 mM) and 8 µl 0.5 M CAA (final 20 mM) were added and transferred to a Vivacon® 500 10 kDa cutoff filter and centrifuged at 14,000 *g* at 20 °C for 15 min. 200 µl of 50 mM ABC was then added and centrifuged again until half wet. This step was repeated twice. Partial digestion was then performed by adding 50 µl of 50 mM ABC with trypsin 1:2000 (trypsin: histone ratio) for 20 min at room temperature. Flow-through and retent were collected by centrifuging the filter units at 14,000 *g* at 20 °C for 10 min. Filters were then washed with 50 µl of 50 mM ABC and flow through was collected. The partially digested protein samples were later stage-tipped for label-free quantitative proteomics.

All samples were analyzed on a Q-Exactive Plus (Thermo Scientific) mass spectrometer that was coupled to an EASY nLC 1200 UPLC (Thermo Scientific). Peptides were loaded with solvent A (0.1% formic acid in water) onto an in-house packed analytical column (50 cm × 75 µm I.D., filled with 2.7 µm Poroshell EC120 C18, Agilent). Peptides were chromatographically separated at a constant flow rate of 250 nl min⁻¹ using 150 min method: 5–30% solvent B (0.1% formic acid in 80% acetonitrile) within 119 min, 30–50% solvent B within 19 min, followed by washing and column equilibration. The mass spectrometer was operated in data-dependent acquisition mode. The MS1 survey scan was acquired from 300 to 1750 *m/z* at a resolution of 70,000. The top 10 most abundant peptides were subjected to higher collisional dissociation (HCD) fragmentation at a normalized collision energy of 27%. The AGC (automatic gain control) target was set to 5e5 charges. Product ions were detected in the Orbitrap at a resolution of 17,500.

All mass spectrometric raw data were processed with MaxQuant (version 1.5.3.8) using default parameters. Briefly, MS2 spectra were searched against the human Uniprot database with human histone sequences, including a list of common contaminants. False discovery rates (FDRs) on protein and peptide–spectrum match (PSM) level were estimated by the target-decoy approach to 0.01% (Protein FDR) and 0.01% (PSM FDR), respectively. The minimal peptide length was set to 7 amino acids and carbamidomethylation at cysteine residues was considered as a fixed modification. The maximum number of missed cleavages was set to 6. Variable modifications included oxidation (M), acetylation (protein N-term and K), methylation (KR) and diglycine (signature of ubiquitination) (K).

**Re-immunoprecipitation of histone H3**. Before immunoprecipitation, HEK293 cells were treated with 5 µM MG-132 for 16 h. Then, cells were lysed in modified RIPA buffer (50 mM Tris-HCl (pH 7.4), 150 mM NaCl, 0.25% sodium deoxycholate, 1% NP40, 1 mM PMSF, 1 mM EDTA, 1 mM NaF, 1 mM Na3VO) supplemented with protease inhibitor cocktail. Lysates were homogenized through a syringe needle (27 G) and centrifuged at 13,000 *g* for 15 min at 4 °C. The supernatant was collected and protein concentration was determined. Approximately 400 µg of protein were used for immunoprecipitation. After pre-clearing of the supernatant with Protein A agarose beads (Pierce), the samples were incubated overnight with 2 µg histone H3 antibody (Cell Signaling, #2650) at 4 °C. As a negative control, the same amount of protein was incubated with 2 µg anti-FLAG antibody (SIGMA, F7425) in parallel. Subsequently, samples were incubated with 30 µl of Protein A beads for 1 h at room temperature. Then, samples were centrifuged for 5 min at 5000 *g* and the pellet was washed three times with RIPA buffer. To elute histone H3, the beads were incubated with 50 µL 0.2 M glycine pH 2.6 (1:1) for 10 min with frequent agitation. The eluate was collected after centrifugation and immediately neutralized by adding an equal volume of Tris pH 8.0. Then, samples were diluted with RIPA buffer and used for re-immunoprecipitation with 2 µg anti-histone H3 or anti-FLAG control antibody. Subsequently, samples were incubated with 30 µl of Protein A beads for 1 h at room temperature. After this incubation, samples were centrifuged for 5 min at 5000 *g* and the pellet was washed three times with RIPA buffer. For the second elution, the beads were incubated with Laemmli Buffer, boiled for 5 min and centrifuged at 13,000 *g* for 5 min. The supernatant was then loaded onto a sodium dodecyl sulfate–polyacrylamide gel electrophoresis (SDS–PAGE) gel for western blot analysis.

**Chromatin Immunoprecipitation-sequencing (ChIP-seq)**. ChIP-seq experiments were performed following the protocol described in ref. [12]. Cells were crosslinked with 1% formaldehyde for 10 min at room temperature. Crosslinked cells were quenched with 0.125 M glycine for 10 min at room temperature and scraped/ transferred to a 15 ml conical tube on ice. Cells were then centrifuged for 5 min at 4 °C followed by two washing steps with 5 ml PBS 1×/1 mM PMSF[66]. H9 hESCs were resuspended sequentially in three different lysis buffers (lysis buffer 1: 50 mM Hepes, 140 mM NaCL, 10% glycerol, 1 mM EDTA, 0.25% TX-100, 0.5% NP-40, and protease inhibitor (Roche); lysis buffer 2: 10 mM Tris, 200 mM NaCL, 0.5 mM EGTA, 1 mM EDTA; lysis buffer 3: 10 mM Tris, 100 mM NaCL, 0.5 mM EGTA, 1 mM EDTA, 0.1% Na-Deoxycholate, 0.5% N-Lauroylsarcosine). Chromatin was sonicated for 20 cycles (30 s on, 45 s off) using Bioruptor (Diagenode). After sonication, the material was centrifuged at 16000 g for 3 min at 4 °C, with the supernatant representing the sonicated chromatin. 75 µl was not subject to immunoprecipitation and was used as total input control for the ChIP reactions.

750 µl were incubated with 10 µg of anti-H3K9me3 antibody (Abcam, #8898, reported suitable for ChIP) overnight at 4 °C. On day 2, magnetic Dynabeads G (Thermofisher) at 10x volume of H3K9me3 antibody were aliquoted into a new microtube. Magnetic beads were washed five times with cold RIPA wash buffer (50 mM Hepes, 1 mM EDTA, 500 mM LiCl, 1% NP-40, 0.7% Na-Deoxycholate). Then, beads were washed once with 1 ml TE + 50 mM NaCl on ice and samples were centrifuged at 950 g for 3 min at 4 °C. After removing all liquid from beads, elution buffer (50 mM Tris, 10 mM EDTA, 1% SDS) was added for 15 min at 65 °C. Finally, beads were centrifuged for 1 min at 1600 g and placed on magnet-holder to settle and the supernatant was transferred into a new tube. To reverse crosslinking, 3x volume of the elution buffer was added to the input and incubated together with the ChIP sample at 65 °C overnight. On day 3, 1x volume of TE buffer and 0.2 mg ml$^{-1}$ RNase were added and incubated for 1 h at 37 °C. To digest proteins, 0.2 mg ml$^{-1}$ Proteinase K was added and incubated for 2 h at 55 °C. Then, DNA was phenol-chloroform extracted at room temperature with 1x volume of 25:24:1 phenol-chloroform-isoamyl alcohol and centrifuged for 5 min to separate layers, followed by the addition of 1x volume chloroform. The DNA was transferred to a new tube for precipitation with 1/10 of NaOAc 3 M, 1 µl 20 mg ml$^{-1}$ glycogen, 2x volume of ice-cold ethanol during 30 min at −80 °C. After centrifugation for 30 min at 4 °C, we removed the supernatant was removed and added 0.5 ml ice cold 70% ethanol followed by 5 min centrifugation. After removing the ethanol, the pellet was air-dried at room temperature and resuspended in 40 µl dH$_2$O.

ChIP-seq libraries from ChIP and input DNA samples were prepared as previously described[67]. Libraries from H9 hESCs were sequenced with a 2 × 75 bp read length on Illumina HiSeq4000. For ChIP-seq, two biological replicates from independent experiments were analysed by QuickNGS (Next-Generation Sequencing) pipeline[55]. Quality check of sequencing data were performed with FastQC version 0.10.1 (Babraham Bioinformatics). ChIP-seq sequencing reads were mapped with Burrows-Wheeler Aligner (BWA)[68] to the Homo Sapiens genome (Ensembl database 87). For peak calling, the resulting Binary Alignment/Map (BAM) files were analyzed with MACS2 version 2.0.10[69]. The results comprise lists of statistically significant peaks compared with the respective input DNA controls. QuickNGS pipeline identifies all genes which are 10,000 bp up- or downstream from the MACS2 peaks. To identify differential read-enriched peak regions from ChIP-seq data between different conditions, we used bdgdiff module of MACS2. Data was uploaded into MySQL database. In addition, QuickNGS provides password-protected track hubs for the UCSC Genome Browser with hyperlinks for visualization.

**Western blot**. hESCs and HEK293 cells were scraped from plates and lysed in protein cell lysis buffer (10 mM Tris-HCl, pH 7.4, 150 mM NaCl, 10 mM EDTA, 1% Triton X-100, 50 mM NaF, 0.1% SDS, 20 µg ml$^{-1}$ Aprotinin, 2 mM sodium orthovanadate, 1 mM phenylmethylsulphonyl fluoride and protease inhibitor) by incubating samples for 10 min on ice and homogenization through 27-G syringe needle. Then, cell lysates were centrifuged at 10,000 g for 10 min at 4 °C and the supernatant was collected. Nematodes were lysed in protein lysis buffer (50 mM Tris-HCl, pH 7.8, 150 mM NaCl, 0.25% sodium deoxycholate, 1 mM EDTA and protease inhibitor) using a Precellys 24 homogenizer. Worm lysates were centrifuged at 10,000 rpm for 10 min at 4 °C and the supernatant was collected.

Protein concentrations were determined with BCA protein assay (Thermoscientific, Germany). Total protein was separated by SDS–PAGE, transferred to nitrocellulose membranes (Millipore, Germany) and immunoblotted. Western blot analysis was performed with anti-UBE2K (Cell Signaling, #8226, 1:1,000), anti-OCT4 (Stem Cell Technologies, #60093, 1:500), anti-SOX2 (Abcam, #97959, 1:1,000), anti-PAX6 (Stem Cell Technologies, #60094, 1:200), anti-Nestin (Stem Cell Technologies, #60091, 1:1,000), anti-MAP2 (Sigma, #1406, 1:1,000), anti-polyubiquitinylated conjugates (Enzo, PW8805-0500, 1:1,000), anti-ubiquitin (Merck Millipore, # 05-944, clone P4D1-A11, 1:1000), anti-H3K9me3 (Abcam, #8898, 1:1,000), anti-Histone H3 (Cell Signaling, #2650, 1:10,000), anti-H3K9me1 (Cell Signaling, #1418, 1:1,000), anti-H3K9me2 (Cell Signaling, #4658, 1:1,000), anti-H3K4me3 (Active Motif, #39916, 1:1,000), anti-H3K27me3 (Active Motif, #39155, 1:1,000), anti-H3K27ac (Active Motif, #39933, 1:1,000), anti-HTT (Cell Signaling, #5656, 1:1,000), anti-SETDB1 (Abcam, #107225, 1:500), anti-p53 (Cell Signaling, #9282, 1:2,000), anti-Histone H1 (Merck, 05-457, 1:1,000), anti-PSMD11 (Abcam, #99413, 1:1,000), anti-ß-actin (Abcam, #8226, 1:1,000) and α-tubulin (Sigma, T6199, 1:5,000).

**In vitro ubiquitination assays**. A concentration of 10 µg of purified human recombinant protein 6xHis::H3F3A (Prospec, PRO-1452) was mixed with 25 ng of E1 activating enzyme (Enzo Life Sciences, BML-UW9410-0050), 400 ng of UBE2K/E2-25K (R&D systems, E2-603), 2 µg of FLAG::ubiquitin (Sigma-Aldrich, U6253), ATP regeneration solution (Enzo Life Sciences, BML-EW9810-0100) and Ubiquitin Conjugation Reaction Buffer (Enzo Life Sciences, BML-KW9885-0001). When indicated in the corresponding figures, we also added 1 µg of hESC lysate or 0.8 µg of recombinant E3 enzymes RNF2 or RNF138. Samples were incubated at 30 °C for 1 h. The reaction was terminated by boiling for 5 min with SDS-sample buffer, and resolved by SDS-PAGE followed by immunoblotting using anti-6xHis antibody (QIAGEN, #34660, 1:1000) to monitor ubiquitination of histone H3F3A. RNF2 cloned into pRK5-FLAG plasmid was a gift from J. Zhang[70]. RNF138 cloned into the eGFP-C1 vector was a gift from M.J. Hendzel[71].

**26S proteasome fluorogenic peptidase assays**. In vitro assay of proteasome activity was performed as previously described[72]. Cells were collected in proteasome activity assay buffer (50 mM Tris-HCl, pH7.5, 250 mM sucrose, 5 mM MgCl2, 2 mM ATP, 0.5 mM EDTA, and 1 mM dithiothreitol) and lysed by passing 10 times through a 27 G needle attached to a 1 ml syringe needle. Lysates were centrifuged at 10,000 g for 10 min at 4 °C. 25 µg of total protein were transferred to a 96-well microtiter plate (BD Falcon) and incubated with Suc-Leu-Leu-Val-Tyr-AMC (Enzo), a fluorogenic proteasome substrate to measure chymotrypsin-like activity. Fluorescence (380 nm excitation, 460 nm emission) was monitored on a microplate fluorometer (EnSpire, Perkin Elmer) every 5 min for 1 h at 37 °C.

**Immunocytochemistry**. Immunocytochemistry experiments were performed as we described in ref. [63]. Cells were fixed with paraformaldehyde (4% in PBS) for 30 min, followed by permeabilization (0.2 % Triton X-100 in PBS for 10 min) and blocking (3% BSA in 0.2% Triton X-100 in PBS for 10 min). Cells were incubated with primary antibody for 2 h at room temperature (Rabbit anti-H3K9me3 (Abcam, #8898, 1:500), Rabbit anti-PAX6 (Stem Cell Technologies, #60094, 1:300), Mouse anti-OCT4 (Stem Cell Technologies, #60093, 1:200), Mouse anti-Nestin (Stem Cell Technologies, #60091, 1:500), Rabbit anti-SOX1 (Stem Cell Technologies, #60095, 1:100), Mouse anti-MAP2 (Sigma, #1406, 1:200), Rabbit anti-H3K9me3 (Abcam, #8898, 1:500)). Cells were then washed with 0.2% Triton-X/PBS and incubated with secondary antibody Alexa Fluor 488 goat anti-mouse (ThermoFisher Scientific, A-11029, 1:500), Alexa Fluor 568 goat anti-rabbit (ThermoFisher Scientific, A-11011, 1:500), and 2 µg ml$^{-1}$ Hoechst 33342 (Life Technologies, #1656104) for 1 h at room temperature. 0.2% Triton-X/PBS and distilled water wash were followed before we mounted the cover slips.

**RNA isolation and quantitative RT–PCR**. RNA extraction and quantitative RT–PCR were performed as described in ref. [63]. We extracted RNA using RNAbee (Tel-Test Inc.) and generated cDNA using qScript Flex cDNA synthesis kit (Quantabio). SybrGreen qPCR experiments were performed with a 1:20 dilution of cDNA using a CFC384 Real-Time System (Bio-Rad) following the manufacturer's instructions. Data were analysed with the comparative 2ΔΔ$^C$t method using the geometric mean of *ACTB* and *GAPDH* as housekeeping genes. Supplementary Data 6 contains the sequences of all the primers used for this assay.

**C. elegans strains and maintenance**. *C. elegans* strains were grown and maintained on standard nematode growth media seeded with *E. coli* (OP50)[73]. Wild-type (N2), SS104 (*glp-4(bn2)*I), CF512 (*fer-15(b26)*II;*fem-1(hc17)*IV), MT13293 (*met-2(n4256)*III) were provided by the *Caenorhabditis* Genetics Center (CGC) (University of Minnesota), which is supported by the NIH Office of Research Infrastructure Programs (P40 OD010440).

For western blot experiments, synchronized animals were raised and fed OP50 *E. coli* at 25 °C until day 1 of adulthood. Then, worms were transferred onto plates with *E. coli* (HT115) containing either empty control vector (L4440) or expressing double-stranded RNAi. *ubc-20* RNAi and *ubc-22* RNAi constructs were obtained from the Vidal RNAi library and sequence verified. Wild-type and MT13293 worms were treated with 100 µg ml$^{-1}$ 5-fluoro-2′deoxyuridine (FUdR) to inhibit the proliferation of progeny.

**Statistics and reproducibility**. In each independent experiment using mammalian cells, biological replicates/wells were averaged for every condition. Then, the data of different independent experiments were averaged. Finally, we compared the average across conditions/groups. For *C. elegans* experiments, sample size determination was done according to standard *C. elegans* approaches. Sample sizes are indicated in the corresponding figure legends and supplementary information. Statistics were derived from at least $n = 3$. No data were excluded from the analyses. Statistical comparisons of western blot quantifications and qPCR data were performed with two-tailed Student's *t* test for unpaired samples using GraphPad Prism 6.0. Statistically significant differences from proteomics data were determined with Perseus (v. 1.5.2.4)[62] after correction for multiple testing following the Benjamini–Hochberg procedure, which calculates false discovery rate (FDR) adjusted *P* values. Analysis of enriched Gene Ontology Biological Processes (GOBPs) were performed using Database for Annotation Visualization and Integrated Discovery (DAVID)[74], that provides statistical comparisons with a modified Fisher's exact test (EASE score). We used MACS2 to calculate fold changes and *P* values from ChIP-seq data, allowing for the identification of statistically significant differential enriched-peak regions between conditions[69]. For RNA-sequencing data, gene expression for the individual samples was calculated with Cufflinks2 as FPKMs[57,58] and fold-change and *P* values were calculated with DEseq2[59].

**Reporting summary**. Further information on research design is available in the Nature Research Reporting Summary linked to this article.

## Data availability

RNA-seq and ChIP-seq data have been deposited in Gene Expression Omnibus (GEO) under the accession code GSE146704. The mass spectrometry proteomics data showed in Supplementary Data 2 and Supplementary Fig. 12 have been deposited to the

ProteomeXchange Consortium via the PRIDE[75] partner repository with the dataset identifiers PXD018625 and PXD018621, respectively. Uncropped images of western blots are presented in Supplementary Data 7. All source data underlying the graphs presented in the main figures can be found as Supplementary Data 8. All the other data are also available from the corresponding author upon request (DV).

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

## Acknowledgements

The Deutsche Forschungsgemeinschaft (DFG) (VI742/1-2 and Germany's Excellence Strategy-CECAD, EXC 2030-390661388), the Else Kröner-Fresenius-Stiftung (2015_A118) and the European Research Council (ERC Starting Grant-677427 Stem-Proteostasis) supported this research. This work was also supported by ERC Consolidator Grant-616499 to T.H., the Foundation for Polish Science co-financed by the European Union under the European Regional Development Fund (POIR.04.04.00-00-5EAB/18-00) to W.P., and the Polish National Science Center (UMO-2016/23/B/NZ3/00753) to A.D. We thank L. Kurian for helpful discussions and comments on the manuscript. We thank the CECAD Proteomics Facility and the Cologne Center for Genomics (CCG) for their contribution and advice in proteomics and RNA sequencing experiments, respectively.

## Author contributions

A.F. and D.I. performed most of the cell experiments, data analysis and interpretation through discussions with D.V. A.N. and C.S. carried out *C. elegans* experiments. M.M.R. performed analysis of proteomics data. A.D. and W.P. contributed to in vitro ubiquitination assays. O.L. performed FASP experiments. V.S.-G. contributed to analysis of ChIP-seq data. P.W. performed analysis of ChIP-seq and RNA-seq data. W.P., T.H. and A.R.-I provided critical advice for proteasome and ChIP-seq experiments. D.V. planned and supervised the project. The manuscript was written by D.V. All the authors discussed the results and commented on the manuscript.

## Competing interests

The authors declare no competing interests.
