## [Peer Review File · Communications Biology]

Reviewers' comments:

Reviewer #1 (Remarks to the Author):

In this manuscript, Fatima et al showed that hESCs exhibit high levels of UBE2K, a ubiquitin-conjugating enzyme. They demonstrated that the reduced expression of UBE2K increases the levels of both total H3 and H3K9 trimethylation, and impairs the ability of hESCs to differentiate into neural progenitors with neurogenic properties. They claimed that UBE2K promotes polyubiquitination and proteasomal degradation of histone H3 in human cells. The understanding of mechanisms underlying histone degradation is of interest to the fields of epigenetic regulation of gene expression. But, the evidence presented in this manuscript is far to be sufficient to support their claims as detailed below:

1. In Fig. 4, the authors showed downregulation of H3 at protein levels by the reduced expression of UBE2K. It is highly possible that the downregulation of H3 was due to the reduced transcription, since UBE2K apparently affects transcription as they showed in Fig. 2.
2. In Fig. 6c, the in vitro ubiquitination assay clearly showed that UBE2K supports monoubiquitination of H3, instead of polyubiquitination. Because monoubiquitination is usually not associated with proteasomal degradation of a substrate, it is apparently incorrect for the authors to claim that UBE2K promotes polyubiquitination and proteasomal degradation of histone H3.
3. In Fig. 6d, the authors showed that the proteasome inhibitor MG132 could increase the levels of H3. Because MG132 can inhibit proteasomal degradation of all substrates, this effect could be indirect. Moreover, except the 26S proteasome (19S-20S), which promotes degradation of polyubiquitinated substrates, there are at least two other types of proteasomes, including REG γ /PA28 γ -20S and PA200-20S proteasomes, which promote the ubiquitin-independent degradation. MG132 can inhibit all types of proteasomes by targeting their catalytic subunits in the 20S particle. Thus, the results in Fig. 6d cannot support their claim for a polyubiquitin-dependent degradation of H3. To valid this claim, an in-vitro degradation assay for H3 with the different types of proteasomes should be performed to show whether the 26S proteasome can promote degradation of polyubiquitinated H3 directly. In addition, mutant cells for different types of proteasome activators should be included to test this issue.
4. The results in Figs. 6e and 6f could not show the increase in polyubiquitination of H3 by addition of UBE2K, even though they claimed so. Particularly, the smeared bands for polyubiquitinated proteins in Fig. 6f were not much different. In addition, because other ubiquitinated proteins might also associate with H3 or anti-H3 antibodies, a re-IP assay by anti-H3 is essential to exclude the contaminated proteins following the first IP and the dissociation of the protein interaction in the SDS buffer at 97 °C.

Reviewer #2 (Remarks to the Author):

The authors identify UBE2K as a novel regulator of neural differentiation and H3 and H3K9me3 levels in ESCs. They demonstrate the UBE2K expression is high in ES and iPS cells and is reduced upon differentiation. Using shRNAs, they go on to show that reducing UBE2K levels does not change expression of pluripotency genes, but instead reduces the ability of ES cells to fully differentiate along the neuronal lineage. To address the mechanism by which UBE2K acts, the group examines the effect of loss of UBE2K on H3 and H3K9me3 levels, finding that they are increased upon UBE2K knockdown. This increase correlates with reduced induction of neuronal gene expression at later stages of differentiation. Finally, the authors find that UBE2K binds H3 to modulate its polyubiquitination, and show that in *C. elegans*, the nematode ortholog of UBE2K modulates H3 and H3K9me3 levels in the germline.

Overall, this study is of broad interest to the field, as it proposes a new mechanism of regulation of

neuronal differentiation through the modulation of histone levels, and a specific mechanism of H3K9me3 regulation. However, the manuscript in its current form falls short of strongly supporting the proposed mechanism. Without additional experiments addressing specificity, the model proposed by the authors is not fully supported. Specific suggestions are listed below.

1. Regarding the specificity of the effect on histone marks, it appears the authors only tested trimethylation of H3K9. It is important and quite straightforward to test other histone modifications, including H3K4me3, H3K4me1, H3K27me3, H3K27ac. If other marks are affected, this may change the proposed model.

2. The legend title for Figure 4 and title for the corresponding text section are confusing. "Impairs" suggests that the H3 and H3K9me3 levels are reduced upon loss of UBE2K.

3. In the ChIPseq section, the authors state that most peaks "were not associated with genes". In the methods, it states that peaks were assigned to genes using a 2kb window. This is quite narrow and it is likely that the authors are missing a number of genes using this window. At the very least, it would be useful to also report the genes using a broader window, otherwise a number of interesting genes may be missed in the analysis.

4. Additional controls should be performed to test the specificity of UBE2K for H3. The amount of H3 that is ubiquitinated in 6b is modest. If UBE2K is a promiscuous enzyme, it is possible that it has a number of targets that would be modified in this assay. It would be helpful to include negative controls in this assay (e.g. other histones) as well as a positive control as a reference for the level of activity on a known target (BRCA1?).

5. All experiments in Fig 6 use overexpression. Does knockdown of UBE2K result in reduced polyubiquitination of H3 in ES cells?

Reviewer #3 (Remarks to the Author):

The manuscript by Fatima et al describes a novel role for ubiquitin-conjugating enzyme UBE2K in maintaining the neurogenic capability of hESCs. The authors discovered that expression of UBE2K is different between hESCs and neural progenitor cells (NPCs) or neuronal counterparts derived from hESCs. Knocking down of UBE2K does not affect the status as well as neural lineage differentiation potential of hESCs, but blocks terminal differentiation of hESCs into neurons. Meanwhile, the authors tested the expression of H3K9me3 after loss of UBE2K since H3K9me3 up-regulated cells show a similar phenomenon regarding differentiation based on a previous report from the same research group. Increase of H3K9me3 and histone 3 was detected in UBE2K-deleted cells which show an enhanced binding of H3K9me3 to neural genes and fail to induce expression of these genes during neuronal differentiation. Molecularly, the authors proved that UBE2K interacted, ubiquitinated H3 and destructed both H3 and H3K9me3. This function of UBE2K is potential interesting. However, I have few concerns:

1. Whether H3K9me3-mediated suppression of pro-neural and neuronal genes is response for neuronal differentiation in UBE2K-depleted cells is not very convincing although phenotype correlated well. Since most cellular phenomenon and effect of the expression on H3K9me3-marked genes appease only between 10~20 days of neural differentiation, elimination H3K9me3 by knockdown of the trimethylase such as SETDB2 or by reconstitution of UBE2K in NPCs prior terminal differentiation should rescue the phenotype and bring these two aspects together.

2. On the destruction of H3 and H3K9me3. Since pan-H3 is dysregulated after deletion of UBE2K, does H3 express in a high level in NPCs where UBE2K is low as compared to hESCs? The authors claim that dysregulation of H3K9me3 is more significant compared to H3 after disruption of UBE2K. Why and how this happens? First, other modifications, such as H3K9me1 and H3K9me2, should be analyzed upon UBE2K to rule out the specific to H3K9me3. An anti-H3K9me3 analysis in Fig 6a could give a direct evidence to see whether UBE2K interacts more strongly to H3K9me3 than other form of H3. High level of H3K9me3 may due to change of activity of H3K9me3 trimethylase. It is reported that UBE2K interacts with Huntingtin (HTT) (Michael A. Kalchman, et al., 1996, JBC) which play role in regulation the activity of the H3K9me3 trimethylase SETDB1 (Irmak D, et al., 2018). The authors should show the effect of UBE2K on trimethylase SETDB1 and/or SETDB1 by western blotting.

3. Another confused points: Map2 is a neuronal marker and expresses similar between NT-shRNA and UBE2K-shRNA treated cells in 10 days of neural differentiation (Fig 3C), why it lost its expression in 20days in shUBE2K cells (Fig 3E)? Similar issue to Tju1 between Fig 3g and Fig 3k. In the other hand, Nestin is thought to be a NPCs marker and why it is detected in NT-shRNA treated cells in 20days of neural differentiation (Fig 3E)?

4. The quantitated present of dysregulation of H3 and H3K9me3 (Fig 4a-b,6d and 7) are misleading. It seems the relative level of H3K9me3 is higher than H3 itself in UBE2K deficient cells. Suggest to present the ratio (H3K9me3/H3) change among different shRNA treatment. Blotting with anti-body to UBE2K in many blots of Fig 6 should be included; anti-RNF138, RNF2 should provided in Fig 6C.

Referee expertise:

Referee #1: Histone degradation during cell division

Referee #2: Transcriptional mechanisms underlying neural stem cell homeostasis

Referee #3: Proteasomal degradation in neurogenesis

Reviewers' comments:

Reviewer #1 (Remarks to the Author):

In this manuscript, Fatima et al showed that hESCs exhibit high levels of UBE2K, a ubiquitin-conjugating enzyme. They demonstrated that the reduced expression of UBE2K increases the levels of both total H3 and H3K9 trimethylation, and impairs the ability of hESCs to differentiate into neural progenitors with neurogenic properties. They claimed that UBE2K promotes polyubiquitination and proteasomal degradation of histone H3 in human cells. The understanding of mechanisms underlying histone degradation is of interest to the fields of epigenetic regulation of gene expression. But, the evidence presented in this manuscript is far to be sufficient to support their claims as detailed below:

1. In Fig. 4, the authors showed downregulation of H3 at protein levels by the reduced expression of UBE2K. It is highly possible that the downregulation of H3 was due to the reduced transcription, since UBE2K apparently affects transcription as they showed in Fig. 2.

*Reviewer #1 is absolutely right and analysis of H3 transcript levels is needed to support the main conclusions of our manuscript. We have now assessed the mRNA levels of histone H3.1, H3.2 and H3.3 variants in hESCs and their corresponding NPC counterparts, and found no significant differences upon UBE2K knockdown (please see **Figure 4c** and **Supplementary Figures 10a, b**). The text now says: "In addition, loss of UBE2K was sufficient to increase the total protein levels of histone H3 in hESCs (**Fig. 4b**). On the contrary, we did not observe interaction of UBE2K with histone H1 or changes in H1 protein levels upon UBE2K shRNA (**Fig. 4a, b**). Since UBE2K knockdown did not affect the transcripts amounts of H3 (**Fig. 4c**), our results suggest a role of UBE2K in the regulation of H3 protein levels". (...) "The increase in total H3 levels of NPCs derived from UBE2K shRNA hESCs was not associated with changes in the amounts of histone H3 transcripts (**Supplementary Fig. 10a, b**)".*

2. In Fig. 6c, the in vitro ubiquitination assay clearly showed that UBE2K supports monoubiquitination of H3, instead of polyubiquitination. Because monoubiquitination is usually not associated with proteasomal degradation of a substrate, it is apparently incorrect for the authors to claim that UBE2K promotes polyubiquitination and proteasomal degradation of histone H3.

We agree with Reviewer #1 that the data presented in Figure 6c of our first submission only supported monoubiquitination. We have now repeated these experiments to acquire data from higher molecular weights and obtained exciting results supporting polyubiquitination of H3 (please see **Figure 9a**). In particular, we found that UBE2K in combination with RNF138 not only induces monoubiquitination of H3 but also (to a lesser extent) polyubiquitination. In contrast, UBE2K in combination with RNF2 did not have these effects. The text now says: "Since UBE2K interacts with histone H3 (**Fig. 4a**), we assessed whether UBE2K modulates its ubiquitination by in vitro ubiquitination assays. For this purpose, we used recombinant UBE2K in combination with either RNF2 or RNF138, the main interacting E3 enzymes of UBE2K³¹⁻³⁵. Whereas the combination of UBE2K with RNF2 did not efficiently promote ubiquitination of H3, UBE2K-RNF138 not only induced robust monoubiquitination of H3 but also polyubiquitination (**Fig. 9a**). Likewise, UBE2K-RNF138 induced ubiquitination of p53 (**Fig. 9b**), a known target of UBE2K³⁶. On the contrary, we did not detect ubiquitination of histone H1 under these conditions (**Fig. 9c**).

3. In Fig. 6d, the authors showed that the proteasome inhibitor MG132 could increase the levels of H3. Because MG132 can inhibit proteasomal degradation of all substrates, this effect could be indirect. Moreover, except the 26S proteasome (19S-20S), which promotes degradation of polyubiquitinated substrates, there are at least two other types of proteasomes, including REGγ/PA28γ-20S and PA200-20S proteasomes, which promote the ubiquitin-independent degradation. MG132 can inhibit all types of proteasomes by targeting their catalytic subunits in the 20S particle. Thus, the results in Fig. 6d cannot support their claim for a polyubiquitin-dependent degradation of H3. To valid this claim, an in-vitro degradation assay for H3 with the different types of proteasomes should be performed to show whether the 26S proteasome can promote degradation of polyubiquitinated H3 directly. In addition, mutant cells for different types of proteasome activators should be included to test this issue.

As Reviewer #1 pointed out, MG-132 is a potent inhibitor of proteasome activity which affects degradation of many substrate proteins. While it is a promising result the detected increase in histone H3 triggered by MG-132 (now **Figure 9d**), the effects of this treatment could be indirect as Reviewer #1 indicated. However, it is important to note that in the same experiment we also show data that rule out indirect effects:

- 1- Overexpressing UBE2K, a component of the ubiquitin-proteasome system that interacts (**Figure 4a**) and promotes ubiquitination of histone H3 (**Figure 9a**), results in lower levels of H3.
- 2- MG-132 antagonizes the effects of UBE2K overexpression on the amount of histone H3 (**Figure 9d**).

We believe that these experiments provide supporting data of a specific role of the E2 enzyme UBE2K in the regulation of histone H3 levels by the proteasome. Since MG-132 can inhibit different types of proteasomes, we agree with Reviewer #1 that it

is important to define which type of proteasome is involved in H3 degradation to demonstrate the link between UBE2K and ubiquitin-mediated proteolysis of H3. The *in vitro* degradation assay proposed by Reviewer #1 could provide information on the correlation between histone H3 ubiquitination and recognition of this modification by isolated proteasomes, but it is technically challenging. For instance, this experiment would require first an *in vitro* ubiquitination assay of H3 followed by *in vitro* degradation assays using distinct types of purified proteasomes, which are inherently labile and, thus, their activity and assembly can be easily affected by the conditions of the ubiquitination assay. Because the physiological substrates of 26S proteasomes are usually polyubiquitinated proteins, we should compare the ability of distinct isolated proteasomes to reduce the levels of ubiquitinated H3 after the *in vitro* ubiquitination assay. However, only part of the recombinant H3 is ubiquitinated in the *in vitro* assay and it would be challenging to conclude whether a potential decrease in the levels of polyubiquitinated H3 is due to the deubiquitination activity of 19S components or degradation. Therefore, it appears out of the scope of this work to address the activities of different proteasomal complexes *in vitro*, which might not necessarily reflect the *in vivo* regulation of histone H3 degradation. For this reason, we followed the Reviewer's advice to test mutant cell models for different types of proteasome activators. We have now generated stable cell lines with a knockdown for key regulators of the distinct proteasome assemblies indicated by the Reviewer. Then, we analyzed the impact on proteasome activity, H3 degradation and UBE2K-dependent changes in histone H3 levels (please see **Figures 9e-h**). These new data underlined the key role of UBE2K in ubiquitin-mediated degradation of histone H3. We are thankful to Reviewer #1 for this suggestion, as it significantly strengthened our conclusions. The text now says: "One step further was to assess whether UBE2K modulates histone H3 levels through the ubiquitin-proteasome system. Whereas ectopic expression of UBE2K did not affect global proteasome activity (**Supplementary Fig. 13**), it reduced total H3 levels and H3K9me3 in human HEK293 cells (**Fig. 9d**). Notably, the treatment with the proteasome inhibitor MG-132 diminished the effects on histone H3 induced by UBE2K overexpression (**Fig. 9d and Supplementary Fig. 13**), indicating that UBE2K determines proteasomal degradation of H3. Active proteasomes exist in distinct forms, but its major assembly is formed through the interaction of the catalytic 20S core with the 19S regulatory particle, generating 26S proteasome complexes that degrade polyubiquitinated proteins¹⁴. The 19S subunit PSMD11 is a crucial regulator of 26S proteasome assembly and activity^{20,21,37}. Besides 19S, the 20S catalytic core can also be activated by other regulatory particles such as PA200 (PSME4) or PA28γ (PSME3), which promote ubiquitin-independent proteolytic degradation¹⁴. Since proteasome inhibitors such as MG-132 target the catalytic subunits of the 20S particle, they can inhibit all types of proteasomes. To determine which is the primary type of proteasome involved in histone H3 degradation, we knocked down specific regulators of distinct assemblies in HEK293 cells (i.e., PSMD11, PSME3 and PSME3). Since PSMD11 is essential for cell viability^{21,37}, we could only obtain a 40% knockdown for this 26S regulator which resulted in approximately a 24% decrease of total proteasome activity (**Fig. 9e, f**). Likewise, knockdown of PSME4/PA200 induced a similar downregulation in proteasome activity (**Fig. 9e, f**). However, a potent knockdown of PSME3 (>90%)

did not impair proteasome activity, indicating that PA28 γ /20S assemblies do not contribute to proteasome activity in these cells (**Fig. 9e, f**). Notably, knockdown of PSMD11 was sufficient to induce a substantial increase in H3 levels (**Fig. 9g**). On the contrary, loss of either PSME4 or PSME3 did not cause significant changes in total histone H3 levels (**Fig. 9g**). Thus, these data indicate that 26S proteasomes mediate proteolysis of histone H3. Importantly, PSMD11 knockdown diminished the degradation of H3 induced by UBE2K overexpression (**Fig. 9h**)”.

4. The results in Figs. 6e and 6f could not show the increase in polyubiquitination of H3 by addition of UBE2K, even though they claimed so. Particularly, the smeared bands for polyubiquitinated proteins in Fig. 6f were not much different. In addition, because other ubiquitinated proteins might also associate with H3 or anti-H3 antibodies, a re-IP assay by anti-H3 is essential to exclude the contaminated proteins following the first IP and the dissociation of the protein interaction in the SDS buffer at 97 °C.

*This is an important point raised by Reviewer #1 and we have now performed re-IP experiments with anti-H3 antibody (please see **Figure 9i**). The text now says: “Since UBE2K can form chains of at least four Lys-48-linked ubiquitin moieties that trigger protein degradation by the 26S proteasome^{38,39}, we performed immunoprecipitation of H3 and examined its ubiquitination status upon UBE2K overexpression. Prior to immunoprecipitation, we treated the cells with proteasome inhibitor to block the degradation of H3 induced by UBE2K. Under these conditions we immunoprecipitated similar amounts of histone H3 in both control and UBE2K-overexpressing cells, but UBE2K overexpression slightly increased H3 polyubiquitination (**Supplementary Fig. 14a, b**). To further exclude contaminating proteins following the first immunoprecipitation, we performed a re-immunoprecipitation with anti-H3 antibody. This assay supported a potential polyubiquitination of histone H3 induced by UBE2K (**Fig. 9i**)”.*

Reviewer #2 (Remarks to the Author):

The authors identify UBE2K as a novel regulator of neural differentiation and H3 and H3K9me3 levels in ESCs. They demonstrate the UBE2K expression is high in ES and iPS cells and is reduced upon differentiation. Using shRNAs, they go on to show that reducing UBE2K levels does not change expression of pluripotency genes, but instead reduces the ability of ES cells to fully differentiate along the neuronal lineage. To address the mechanism by which UBE2K acts, the group examines the effect of loss of UBE2K on H3 and H3K9me3 levels, finding that they are increased upon UBE2K knockdown. This increase correlates with reduced induction of neuronal gene expression at later stages of differentiation. Finally, the authors find that UBE2K binds H3 to modulate its polyubiquitination, and show that in *C. elegans*, the nematode ortholog of UBE2K modulates H3 and H3K9me3 levels in the germline.

Overall, this study is of broad interest to the field, as it proposes a new mechanism of regulation of neuronal differentiation through the modulation of histone levels, and a specific mechanism of H3K9me3 regulation. However, the manuscript in its current form falls short of strongly supporting the proposed mechanism. Without additional experiments addressing specificity, the model proposed by the authors is not fully supported. Specific suggestions are listed below.

1. Regarding the specificity of the effect on histone marks, it appears the authors only tested trimethylation of H3K9. It is important and quite straightforward to test other histone modifications, including H3K4me3, H3K4me1, H3K27me3, H3K27ac. If other marks are affected, this may change the proposed model.

*We agree with Reviewer #2 that testing other H3 modifications is needed for the conclusions of our manuscript. This was also pointed out by Reviewer #3. We have now examined by Western blot other histone H3 modifications in hESCs upon UBE2K knockdown (please see **Figure 5b**). Similar to total histone H3 protein levels, we found that the global amounts of histone H3 modifications H3K4me3, H3K27me3 and H3K27ac were also increased upon UBE2K shRNA. However, the ratio of these modifications was not significantly enriched when normalized by total histone H3. In contrast, loss of UBE2K not only increased global H3K9me3 levels but also induced the enrichment of H3K9me3 among total histone H3 (**Figures 4d and 5b**). Importantly, we have now performed co-immunoprecipitation experiments that also indicate more interaction of UBE2K with H3K9me3 than H3K4me3, H3K27me3, and H3K27ac (**Figure 5c**). Moreover, we have now examined the levels of H3K9me1 and H3K9me2 upon UBE2K knockdown (**Figure 5a**). We found that the ratio of H3K9me1 and H3K9me2 was diminished or not significantly changed when normalized by total histone H3, respectively. Taken together, these experiments support a certain degree of specificity in the effects of UBE2K over H3K9 trimethylation. The text now says: "Given the robust increase in H3K9me3/H3 ratio induced by loss of UBE2K, we asked whether UBE2K also modulates other H3 modifications. Upon UBE2K knockdown, H3K9me1/H3 and H3K9me2/H3 ratios were diminished and not significantly changed, respectively (**Fig. 5a**). Whereas loss of UBE2K also induced an up-regulation in the global levels of other H3 modifications (i.e., H3K4me3, H3K27me3, H3K27ac), the ratio of these modifications was not significantly enriched when normalized by total H3 amounts (**Fig. 5b**). In addition, co-immunoprecipitation experiments indicate a more robust interaction of UBE2K with H3K9me3 than H3K4me3, H3K27me3, and H3K27ac (**Fig. 5c**). Thus, UBE2K could particularly modulate H3K9 trimethylation".*

Since loss of UBE2K particularly increased the H3K9me3/H3 ratio, we focused on the impact of up-regulated H3K9me3 in neurogenesis from hESCs. Although UBE2K knockdown did not significantly change the ratio of other histone H3 modifications when normalized to total H3, we cannot discard that the increase in the total levels of these modifications also impairs neurogenesis. We have discussed this important point in the Discussion section.

2. The legend title for Figure 4 and title for the corresponding text section are confusing. "Impairs" suggests that the H3 and H3K9me3 levels are reduced upon loss of UBE2K.

We have now replaced "impairs" by "increases" or "up-regulates" in the legend title for Figure 4 as well as the title and text for the corresponding section. Likewise, we have also made this correction in the text section corresponding to the effects of ubc-20 knockdown in C. elegans germline.

3. In the ChIPseq section, the authors state that most peaks "were not associated with genes". In the methods, it states that peaks were assigned to genes using a 2kb window. This is quite narrow and it is likely that the authors are missing a number of genes using this window. At the very least, it would be useful to also report the genes using a broader window, otherwise a number of interesting genes may be missed in the analysis.

We have now replaced this list by a new analysis where the peaks were assigned to genes using a broader window (10 kb up-or downstream from the MACS2 peak). The number of genes increased significantly. For instance, we found an additional potassium voltage-gated channel (KCNA1).

4. Additional controls should be performed to test the specificity of UBE2K for H3. The amount of H3 that is ubiquitinated in 6b is modest. If UBE2K is a promiscuous enzyme, it is possible that it has a number of targets that would be modified in this assay. It would be helpful to include negative controls in this assay (e.g. other histones) as well as a positive control as a reference for the level of activity on a known target (BRCA1?).

*To test the specificity of UBE2K for H3, we have now examined histone H1. We found that UBE2K does not interact with histone H1 in hESCs (please see **Figure 4a**). Moreover, UBE2K knockdown did not change histone H1 levels in hESCs (**Figure 4b**).*

Regarding the in vitro ubiquitination assays using hESC extracts presented in Figure 6b of our first submission, we completely agree with Reviewer #2 that many targets could be modified in this assay given the potential promiscuity of UBE2K and the fact of using hESC extracts. We have now repeated these experiments to acquire data from higher molecular weights in order to assess both monoubiquitination and polyubiquitination. In these new series of experiments, the signal corresponding to monoubiquitination is less intense compared with previous experiments using recombinant UBE2K + hESC extract. Under these conditions, we also observed signal that could correspond to polyUb-H3 (please see Figure below). However, the signal is very modest, making difficult to interpret these results. We also performed in vitro experiments under the same conditions to test ubiquitination of p53, a target of UBE2K (Kikuchi et al, Arterioscler Thromb Vasc Biol, 2000). Since we could not detect ubiquitination of p53 with recombinant UBE2K + hESC extracts (Figure below), we decided to discard the experiments using hESC extracts. Instead, we focused on in

in vitro experiments using recombinant UBE2K in combination with either RNF2 or RNF138, the main interacting E3 enzymes of UBE2K and obtained exciting results supporting polyubiquitination of H3 (please see **Figure 9a**). Whereas the combination of UBE2K with RNF2 did not efficiently promote ubiquitination of H3, UBE2K-RNF138 not only induced robust monoubiquitination of H3 but also polyubiquitination (**Figure 9a**). Likewise, UBE2K-RNF138 induced ubiquitination of p53 (**Figure 9b**). On the contrary, we did not detect ubiquitination of histone H1 under these conditions (**Figure 9c**). Thus, *in vitro* experiments indicate that UBE2K can promote ubiquitination of H3.

Figure: *In vitro* ubiquitination assays with recombinant UBE2K and hESC extracts. **a**, *In vitro* ubiquitination assay of 6xHis-tagged histone H3F3A with hESC extract followed by electrophoresis and immunoblotting with antibody to 6xHis. The images are representative of two independent experiments. **b**, *In vitro* ubiquitination assay of recombinant p53 with hESC extract followed by electrophoresis and immunoblotting with antibody to 6xHis. The images are representative of two independent experiments.

5. All experiments in Fig 6 use overexpression. Does knockdown of UBE2K result in reduced polyubiquitination of H3 in ES cells?

This is a very interesting experiment and we were very excited about it. Unfortunately, we found a roadblock that we could not overcome. Since UBE2K knockdown induces >2-fold increase of total H3 levels, this experiment requires the treatment with proteasome inhibitor to block the degradation of H3 induced by UBE2K in control hESCs. Thus, by treating both control and UBE2K shRNA hESCs with proteasome inhibitor we would be able to immunoprecipitate similar amounts of histone

H3 and compare polyubiquitination between the two conditions. However, hESCs are extremely sensitive to proteasome inhibitor when compared with other cells (Vilchez et al, Nature 2012). In our first experiments, the whole culture died and all the cells were detached within 1 hour when we used 5 μ M MG-132, which is the standard concentration in cell culture. We then tried lower concentrations of MG-132. When we decreased the MG-132 concentration almost 100 times, we could observe a few cells which appeared to be alive and remained attached to the plate, but their morphology was very different compared with untreated hESCs. Nevertheless, we lysed these cells for immunoprecipitation but even combining numerous plates we did not obtain enough protein to immunoprecipitate histone H3. We are very sorry because after trying this experiment numerous times, we believe it is technically impossible for us to obtain enough material for immunoprecipitation of hESCs treated with proteasome inhibitor.

Reviewer #3 (Remarks to the Author):

The manuscript by Fatima et al describes a novel role for ubiquitin-conjugating enzyme UBE2K in maintaining the neurogenic capability of hESCs. The authors discovered that expression of UBE2K is different between hESCs and neural progenitor cells (NPCs) or neuronal counterparts derived from hESCs. Knocking down of UBE2K does not affect the status as well as neural lineage differentiation potential of hESCs, but blocks terminal differentiation of hESCs into neurons. Meanwhile, the authors tested the expression of H3K9me3 after loss of UBE2K since H3K9me3 up-regulated cells show a similar phenomenon regarding differentiation based on a previous report from the same research group. Increase of H3K9me3 and histone 3 was detected in UBE2K-deleted cells which show an enhanced binding of H3K9me3 to neural genes and fail to induce expression of these genes during neuronal differentiation. Molecularly, the authors proved that UBE2K interacted, ubiquitinated H3 and destructed both H3 and H3K9me3. This function of UBE2K is potential interesting. However, I have few concerns:

1. Whether H3K9me3-mediated suppression of pro-neural and neuronal genes is response for neuronal differentiation in UBE2K-depleted cells is not very convincing although phenotype correlated well. Since most cellular phenomenon and effect of the expression on H3K9me3-marked genes appease only between 10~20 days of neural differentiation, elimination H3K9me3 by knockdown of the trimethylase such as SETDB2 or by reconstitution of UBE2K in NPCs prior terminal differentiation should rescue the phenotype and bring these two aspects together.

*To perform rescue experiments, we have now knocked down the expression of UBE2K using a shRNA against the 3'UTR region of UBE2K transcript followed by ectopic expression of UBE2K cDNA in hESCs (please see **Figure 8**). In support of our conclusions, we observed that ectopic expression of UBE2K rescues H3K9me3 levels in UBE2K shRNA hESCs (**Figure 8a**). After 20 days on neural induction treatment, ectopic expression of UBE2K rescued the low expression of H3K9me3-enriched neuronal genes (i.e., GBX1, HES6, KCNA3, KCNA5) as well as other neuronal*

markers such as MAP2 and NEFL (**Figures 8b-c**). The text now says: “To further examine the link between UBE2K, H3K9me3 and induction of neuronal genes, we performed rescue experiments. For this purpose, we knocked down the expression of UBE2K using a shRNA against the 3'UTR region of UBE2K transcript and rescue H3K9me3 levels by ectopic expression of UBE2K cDNA in hESCs (**Fig. 8a**). Upon 20 days on neural induction treatment, 3'UTR UBE2K shRNA cells exhibited decreased expression of H3K9me3-marked genes (i.e., GBX1, HES6, KCNA3, KCNA5) when compared with control cells (**Fig. 8b**). Notably, ectopic expression of UBE2K rescued this phenotype (**Fig. 8b**) as well as low expression of other neural and neuronal markers such as MAP2 (**Fig. 8c**). Taken together, our data indicate that UBE2K maintains low levels of H3K9me3 marks in distinct genes, allowing neuronal differentiation of hESCs. Conversely, loss of UBE2K dysregulates H3K9me3 in hESCs, a process that could contribute to altered gene expression upon differentiation and compromised neurogenesis”.

2. On the destruction of H3 and H3K9me3. Since pan-H3 is dysregulated after deletion of UBE2K, does H3 express in a high level in NPCs where UBE2K is low as compared to hESCs? The authors claim that dysregulation of H3K9me3 is more significant compared to H3 after disruption of UBE2K. Why and how this happens? First, other modifications, such as H3K9me1 and H3K9me2, should be analyzed upon UBE2K to rule out the specific to H3K9me3. An anti-H3K9me3 analysis in Fig 6a could give a direct evidence to see whether UBE2K interacts more strongly to H3K9me3 than other form of H3. High level of H3K9me3 may due to change of activity of H3K9me3 trimethylase. It is reported that UBE2K interacts with Huntingtin (HTT) (Michael A. Kalchman, et al., 1996, JBC) which play role in regulation the activity of the H3K9me3 trimethylase SETDB1 (Irmak D, et al., 2018). The authors should show the effect of UBE2K on trimethylase SETDB1 and/or SETDB1 by western blotting.

*Given the links between UBE2K and histone H3 indicated by our results, we agree with Reviewer #3 that it would be interesting to assess changes in histone H3 during neural differentiation of wild-type hESCs. These experiments are presented in **Supplementary Figure 8**. The text now says: “Since UBE2K decreases during differentiation, we asked whether this downregulation correlates with changes in histone H3. Whereas the levels of H3K9me3 dramatically increased upon neural differentiation of control hESCs, we did not observe changes in total H3 levels (**Supplementary Fig. 8**), indicating that other mechanisms could contribute to maintaining physiological amounts of histone H3. However, when UBE2K shRNA hESCs were differentiated into early NPCs, these cells retained the alterations in both total H3 and H3K9 trimethylation induced at the undifferentiated state (**Fig. 6a, b and Supplementary Fig. 9**). The increase in total H3 levels of NPCs derived from UBE2K shRNA hESCs was not associated with changes in the amounts of histone H3 transcripts (**Supplementary Fig. 10a, b**). Altogether, these data suggest that epigenetic changes induced by UBE2K knockdown in hESCs are transmitted to their neural counterparts”.*

Regarding the potential specific effects of UBE2K to H3K9me3, we completely agree with Reviewer #3 that analysing other histone H3 modifications is necessary to support this conclusion. In these lines, Reviewer #2 also suggested similar experiments. We have now examined by Western blot other histone H3 modifications in hESCs upon UBE2K knockdown (please see **Figure 5b**). Similar to total histone H3 protein levels, we found that the global amounts of histone H3 modifications H3K4me3, H3K27me3 and H3K27ac were also increased upon UBE2K shRNA. However, the ratio of these modifications was not significantly enriched when normalized by total histone H3. In contrast, loss of UBE2K not only increased global H3K9me3 levels but also induced the enrichment of H3K9me3 among total histone H3 (**Figures 4d and 5b**). Importantly, we have now performed co-immunoprecipitation experiments that also indicate more interaction of UBE2K with H3K9me3 than H3K4me3, H3K27me3, and H3K27ac (**Figure 5c**). Moreover, we have now examined the levels of H3K9me1 and H3K9me2 upon UBE2K knockdown (**Figure 5a**). We found that the ratio of H3K9me1 and H3K9me2 was diminished or not significantly changed when normalized by total histone H3, respectively. Taken together, these experiments support a certain degree of specificity in the effects of UBE2K over H3K9 trimethylation. The text now says: "Given the robust increase in H3K9me3/H3 ratio induced by loss of UBE2K, we asked whether UBE2K also modulates other H3 modifications. Upon UBE2K knockdown, H3K9me1/H3 and H3K9me2/H3 ratios were diminished and not significantly changed, respectively (**Fig. 5a**). Whereas loss of UBE2K also induced an up-regulation in the global levels of other H3 modifications (i.e., H3K4me3, H3K27me3, H3K27ac), the ratio of these modifications was not significantly enriched when normalized by total H3 amounts (**Fig. 5b**). In addition, co-immunoprecipitation experiments indicate a more robust interaction of UBE2K with H3K9me3 than H3K4me3, H3K27me3, and H3K27ac (**Fig. 5c**). Thus, UBE2K could particularly modulate H3K9 trimethylation".

Since loss of UBE2K particularly increased the H3K9me3/H3 ratio, we focused on the impact of up-regulated H3K9me3 in neurogenesis from hESCs. Although UBE2K knockdown did not significantly change the ratio of other histone H3 modifications when normalized to total H3, we cannot discard that the increase in the total levels of these modifications also impairs neurogenesis. We have discussed this important point in the Discussion section.

We have now performed the experiments suggested by Reviewer #3 to assess a potential link between UBE2K and SETDB1, and obtained exciting data that further support a role of UBE2K in H3K9me3 regulation. First, we confirmed in hESCs that UBE2K interacts with HTT (**Figure 5d**), an inhibitor of SETDB1 activity. Notably, we found that loss of UBE2K reduces HTT levels (**Figure 5e**), a process that could contribute to increased trimethylation of H3K9. In addition, we found that UBE2K also interacts with SETDB1 in hESCs (**Figure 5d**). Remarkably, loss of UBE2K induced a dramatic increase in SETDB1 levels (**Figure 5e**), resulting in a gain of interaction between SETDB1 and histone H3 in hESCs (**Figure 5f**). The Results section now says: "Since HTT inhibits the H3K9 methyltransferase activity of SETDB1 in hESCs¹¹, we first confirmed that UBE2K interacts with HTT in these cells (**Fig. 5d**). Notably, loss of

UBE2K resulted in a downregulation of HTT levels, correlating with higher H3K9 trimethylation (**Fig. 5e**). Besides HTT, we found that UBE2K also interacts with SETDB1 (**Fig. 5d**). Moreover, knockdown of UBE2K dramatically increased the amounts of SETDB1 (**Fig. 5e**). Concomitantly, histone H3 gained interaction with SETDB1 in hESCs, a process that could contribute to H3K9 trimethylation on UBE2K knockdown (**Fig. 5f**). The Discussion section says: “Our data indicate that UBE2K impinges upon H3K9 trimethylation by two mechanisms converging on SETDB1. First, we observe that loss of UBE2K decreases the levels of HTT, an inhibitor of SETDB1 activity. In these lines, our results in *C. elegans* support that H3K9 trimethylation upon loss of UBE2K requires the activity of trimethylases such as SETDB1. Besides changes in HTT, knockdown of UBE2K results in higher SETDB1 protein levels. Given the important role of HTT and SETDB1 in gene expression, it will be fascinating to define how UBE2K regulates their levels. Since UBE2K interacts with SETDB1, an intriguing hypothesis is that UBE2K directly modulates its ubiquitination and proteasomal degradation. Another possibility is that UBE2K modulates SETDB1 levels via its recruitment into histone complexes, preventing its degradation”.

3. Another confused points: Map2 is a neuronal marker and expresses similar between NT-shRNA and UBE2K-shRNA treated cells in 10 days of neural differentiation (Fig 3C), why it lost its expression in 20days in shUBE2K cells (Fig 3E)? Similar issue to Tju1 between Fig 3g and Fig 3k. In the other hand, Nestin is thought to be a NPCs marker and why it is detected in NT-shRNA treated cells in 20days of neural differentiation (Fig 3E)?

We apologize for not making more clear these points in our first submission. For this purpose, we have now included a comparison between 20 days-NPCs with their differentiated neuronal counterparts in Fig. 3k. Second, we have further explained in the main text, figures and methods the distinct differentiation stages assessed in this paper.

Figure 3 presents results from cells at four distinct stages:

- hESCs
- 10 days-NPCs: hESCs were on neural induction treatment for 10 days. At this stage, the cultures consisted mostly of cells that lose the expression of OCT4 but express high levels of PAX6 (**Fig. 3a**), an early NPC marker. These early NPCs (10 days) have a morphology characteristic of NPCs and show induction of NPCs markers such as Nestin or SOX1 when compared with hESCs (**Fig. 3c**). In fact, most of the cells in the culture are Nestin and SOX1-positive (**Supplementary Fig. 2**). However, early NPCs do not have a strong neurogenic potential at this stage, and they did not efficiently generate neurons when directly treated with neuronal differentiation medium. To increase their neurogenic potential, we maintained the cells during a total of 20 days on neural induction medium, a treatment that generated mature NPCs with the ability to differentiate into neurons (see below).

- 20 days-NPCs: hESCs were on neural induction treatment for 20 days. In these mature NPCs, the expression of PAX6 dramatically decreased when compared with 10 day-NPCs (**Fig. 3f**). In contrast, the levels of NPC markers such as Nestin and SOX1 were further increased. Mature NPCs proliferated and maintained their self-renewal ability. In addition, they efficiently differentiated into neurons when treated on neuronal differentiation medium.
- Neurons: Mature NPCs (i.e., 20 days on neural induction medium) were differentiated into neurons using neuronal differentiation media for 1 month.

Our results indicate that loss of UBE2K in hESCs does not affect the first steps of differentiation into early NPCs (10 days on neural induction), but impairs their development into more mature NPCs (20 days on neural induction) and neurogenesis. We believe that the changes observed in MAP2, TUJ1 and Nestin at the distinct differentiation stages support our conclusions, but we totally understand the point raised by Reviewer #3. We have now made more clear our results in the main text and explain in more detail changes in MAP2, TUJ1 and Nestin upon both differentiation and UBE2K shRNA.

Indeed, MAP2 is expressed at low levels in neural progenitors, but its levels increase through the differentiation process into neurons. Thus, MAP2 is usually used as a neuronal marker as indicated by Reviewer #3. However, MAP2 can also be used to distinguish early NPCs (PAX6-positive cells) from mature NPCs, because these cells express higher levels of MAP2 when compared with early NPCs (Carosso et al, JCI Insight 2019). In these lines, our qPCR experiments also show a direct correlation between MAP2 levels and the differentiation stage. In control early NPCs (day 10), we observed higher levels of MAP2 when compared with hESCs (**Fig. 3c**). In control mature NPCs (20 days), there is a further increase in MAP2 levels (**Fig. 3g**). As expected, neurons exhibit a further up-regulation in MAP2 levels compared with 20 days-NPCs (**Fig. 3k**). Similar to the other transcripts tested in early NPCs (PAX6, Nestin and SOX1), loss of UBE2K does not affect MAP2 induction at this stage (**Fig. 3c**). However, loss of UBE2K inhibits its further induction during differentiation into mature NPCs (20 days) and neurons. These results correlate with the impairment in the progressive induction of other genes assessed in mature NPCs and neurons (**Fig. 3g and 3k**).

Similar to MAP2 and other neuronal markers (e.g., AADC, TH+, SYN1, GABAR, DLG4), TUJ1 expression increased in 20 days-NPCs compared with 10-day early NPCs, and was further induced during neuronal differentiation (**Fig. 3g and 3k**). However, UBE2K shRNA does not affect TUJ1 levels in mature NPCs and only appears to block TUJ1 induction during neuronal differentiation (**Fig. 3g and 3k**). Thus, these results indicate a delay on the impact of UBE2K knockdown in TUJ1 levels when compared with other neuronal markers tested. We have now discussed these differences in the Discussion section.

Regarding the expression of Nestin in control cells after 20 days of neural induction, it is important to remark that these cells are essentially still NPCs: e.g., they proliferate and differentiate into neurons. In fact, we observed a higher expression of Nestin and SOX1 in control mature NPCs when compared with control early NPCs

(Fig. 3g). Notably, UBE2K knockdown did not affect the expression of Nestin and SOX1 in early NPCs (**Fig. 3c**), but blocked their further induction during differentiation into mature NPCs (**Fig. 3g**). Altogether, our data indicate that UBE2K shRNA treatment in hESCs does not affect their commitment to a neuroectoderm fate, but impairs their ability to generate mature NPCs with intact neurogenic properties. Our results regarding the link between UBE2K levels and H3K9me3 changes support this hypothesis. In hESCs, knockdown of UBE2K induces H3K9me3 enrichment in genes required for neural and neuronal differentiation (e.g., GBX1, HES6, KCNA3, KCNA5) (**Fig. 7a and Supplementary Data 3**). However, H3K9me3 repressive marks in these genes do not affect their low expression in hESC and early NPCs (**Fig. 7b, c**), but diminish their induction during further differentiation (**Fig. 7e, f**). The H3K9me3-mediated repression of genes such as GBX1, HES6, KCNA3 or KCNA5 could impair differentiation and contribute to the impaired induction of neural (Nestin, SOX1) and neuronal markers (MAP2, AADC, TH+, SYN1, GABAR, DLG4) during differentiation into mature NPCs and neurons. We have now discussed this in more detail in the Discussion section. The text now says: “Importantly, loss of UBE2K not only increases global H3K9 trimethylation but also H3K9me3 enrichment in genes required for neural and neuronal differentiation such as GBX1, HES6 or distinct potassium voltage-gated channels. It is important to note that these changes do not have strong effects on hESCs and their commitment into early NPCs. However, H3K9me3 enrichment in the aforementioned genes diminishes their induction during differentiation into mature NPCs and neurons. Interestingly, a recent study found that low levels of H3K9me3 in hESCs allow for the establishment of compacted heterochromatin in specific protein coding-genes at later stages, a critical process for cell differentiation⁸. Thus, UBE2K could contribute to maintaining low H3K9me3 levels in hESCs, defining cell differentiation potential. In these lines, we observed that loss of UBE2K in hESCs does not affect the first steps of differentiation into early NPCs, but impairs their ability to differentiate into mature NPCs and neurons. We speculate that the H3K9me3-mediated repression of genes such as GBX1, HES6, KCNA3, KCNA5 could contribute to these phenotypes, including the lack of induction in distinct neural and neuronal markers during mid and late stages of differentiation. For instance, UBE2K knockdown does not affect the expression of the neural markers Nestin and SOX1 in early NPCs, but blocks their further induction during differentiation into mature NPCs. When compared with early NPCs, control mature NPCs also have increased levels of all the neuronal markers tested (i.e., AADC, TH+, MAP2, TUJ1, SYN1, NEUN, GABAR, NEFL, NEFM, DLG4). Importantly, loss of UBE2K in hESCs is sufficient to inhibit the induction of these factors during differentiation into mature NPCs, with the exception of TUJ1. Moreover, UBE2K knockdown also blocked the further induction of all the tested neuronal markers during terminal differentiation into neurons, including TUJ1. Despite the delay in TUJ1 alterations, our results indicate that UBE2K knockdown already has deleterious effects during differentiation into mature NPCs”.

4. The quantitated present of dysregulation of H3 and H3K9me3 (Fig 4a-b,6d and 7) are misleading. It seems the relative level of H3K9me3 is higher than H3 itself in UBE2K deficient cells. Suggest to present the ratio (H3K9me3/H3) change among

different shRNA treatment. Blotting with anti-body to UBE2K in many blots of Fig 6 should be included; anti-RNF138,RNF2 should provided in Fig 6C.

*We have now presented H3K9me3/H3 ratios instead of H3K9me3/actin (please see **Figures 4d, 5b, 6a, 8a, 9d, 10a-c, Supplementary Figures 7, 9**). We really appreciate this suggestion because it made more clear our results regarding the effects of UBE2K in H3K9 trimethylation.*

*Figure 6 is now presented as **Figure 9**. We have now included data of UBE2K, RNF2 and RNF138 levels to the in vitro ubiquitination assays (**Figures 9a-c**). Likewise, **Figure 9d** (i.e., UBE2K OE + MG-132 treatment) as well as **Figures 9g, h** (i.e., knockdown of proteasome regulators) contain blots using an anti-UBE2K specific antibody.*

Reviewers' comments:

Reviewer #1 (Remarks to the Author):

My previous concerns on the following critical issues have not been addressed satisfactorily. So, the manuscript is not acceptable for publication.

1. I previously commented that Figure 6c only supported monoubiquitination, rather than polyubiquitination. The authors repeated these experiments and argued that the current results support polyubiquitination of H3 (please see Figure 9a). These repeated data are still not sufficient to support their claims on polyubiquitination. Usually, polyubiquitination of a given protein leads to smeared bands at much higher molecular weights (e.g., >200 kDa), instead of the distinct bands below 50 kDa in the revised Fig. 9a, which can be caused by multi-monoubiquitination. Because monoubiquitination is usually not associated with proteasomal degradation of a substrate, it is important to clarify this issue by using the mutated ubiquitins that cannot form polyubiquitination in the assay.

2. I felt that the results in previous Fig. 6d could not support their claim for a polyubiquitin-dependent degradation of H3, and suggested that the authors should perform an in-vitro degradation assay with the different types of proteasomes. Instead, the authors addressed this issue using shRNA to knockdown the related proteasome activators. Because the 19S subunit PSMD11 is essential for cell viability, the effects of PSMD11 knockdown on H3 levels could be indirectly caused by the UBE2K-mediated monoubiquitination of other substrates. Moreover, the quality of the images in the revised Fig. 9g and 9h is too poor to reflect the difference. For instance, the band for shRNA of PSMD4/PA200 (Fig. 9g) or PSMD11 (Fig. 9h) is much broader than the control band. Thus, the authors should perform an in-vitro degradation assay with the different types of proteasomes, if they prefer to claim that the degradation of H3 is catalyzed by the 26S proteasome, instead of the PA200-proteasome. Otherwise, this conclusion should be revised into that H3 is degraded by proteasomes in general, instead of the 26S proteasome specifically.

Reviewer #2 (Remarks to the Author):

My concerns have been adequately addressed and I support publication in Communications Biology.

Reviewer #3 (Remarks to the Author):

The author addressed fully the questions I raised in last version and now they offered more evidences to support their conclusion. It seems that UBE2K has impact on both protein level of H3 itself and H3K9Me3 modification through ubiquitination and HTT-SETDB1 axis, respectively. However, it is not clear which, ubiquitination-mediated stability of H3 or H3K9Me3, is the key player in regulating the neurogenic potential of the hESCs. Second, what is the relation between H3K9Me3 and UBE2K-mediated ubiquitination of H3. The authors should discuss these points.

Referee expertise:

Referee #1: Histone degradation during cell division

Referee #2: Transcriptional mechanisms underlying neural stem cell homeostasis

Referee #3: Proteasomal degradation in neurogenesis

Reviewers' comments:

Reviewer #1 (Remarks to the Author):

My previous concerns on the following critical issues have not been addressed satisfactorily. So, the manuscript is not acceptable for publication.

We respectfully disagree with Reviewer #1 and we believe that we satisfactorily addressed all of their previous concerns in our first revision with new experiments and supportive data. These data strengthened our conclusions that UBE2K modulates polyubiquitination and proteasomal-mediated degradation of H3, which are the main concerns of Reviewer #1. For instance:

- 1- We presented new in vitro ubiquitination experiments showing higher molecular weights supporting that UBE2K not only induces monoubiquitination but also polyubiquitination of H3 in vitro (**Figure 9a**). Please see a more detail explanation below.*
- 2- Re-IP of H3 followed by western blot with antibody against polyubiquitinated proteins demonstrates that overexpression of UBE2K increases polyubiquitination of H3 in human cells (**Figure 9i**).*
- 3- UBE2K increases H3 protein levels without affecting H3 transcript levels (**Figure 4b, c**).*
- 4- Overexpressing UBE2K, a component of the ubiquitin-proteasome system that interacts (**Figure 4a**) and promotes ubiquitination of histone H3 (**Figure 9a**), results in lower protein levels of H3.*
- 5- Both MG-132 treatment and PSMD11 knockdown antagonize the effects of UBE2K overexpression on the amount of histone H3 (**Figure 9d, 9h**). This is the standard experiment to demonstrate a role of E2/E3 enzymes in proteasomal degradation of a substrate.*

1. I previously commented that Figure 6c only supported monoubiquitination, rather than polyubiquitination. The authors repeated these experiments and argued that the current results support polyubiquitination of H3 (please see Figure 9a). These repeated data are still not sufficient to support their claims on polyubiquitination. Usually, polyubiquitination of a given protein leads to smeared bands at much higher molecular weights (e.g., >200 kDa), instead of the distinct bands below 50 kDa in the revised Fig. 9a, which can be caused by multi-monoubiquitination. Because monoubiquitination is usually not associated with proteasomal degradation of a substrate, it is important to

clarify this issue by using the mutated ubiquitins that cannot form polyubiquitination in the assay.

We previously agreed with the Reviewer #1's comment regarding that the data presented in the *in-vitro* assay of our first submission only supported monoubiquitination of H3 by UBE2K. As explained in our first revision, this was a mistake from our side because we only focused on low molecular weights (up to 35 kDa). We apologize again for it. However, this was already corrected in our first revision by repeating the experiments and acquiring data from higher molecular weights. Indeed, these experiments show that UBE2K in combination with RNF138 induces robust monoubiquitination of H3 (band detected at approx. 25 kDa) *in vitro*. However, we also found ubiquitinated H3 signal at higher molecular weights that could correspond to polyubiquitination events. Reviewer #1 argues that: "Usually, polyubiquitination of a given protein leads to smeared bands at much higher molecular weights (e.g., >200 kDa), instead of the distinct bands below 50 kDa in the revised Fig. 9a, which can be caused by multi-monoubiquitination". First, it is important to note that H3 is a small protein of ~15 kDa, but the Reviewer seems to expect >200 kDa bands if H3 is polyubiquitinated. Even when polyubiquitin chains may also induce some slight changes in motility in the western blot assay that could not completely reflect the real molecular weight, >200 kDa bands would indicate that H3 has extremely long ubiquitin chains (>21 ubiquitin) which it is more improbable than shorter polyubiquitin chains. The Reviewer also mentions that polyubiquitination of a given protein usually leads to smeared bands, but it is important to note that this is not always the case. For instance, UBE2K has been reported to induce polyubiquitination events that do not lead to smeared bands. In the figure 5c (please see below) of Christensen et al (*Nature Structural & Molecular Biology* 2007), the authors showed a UBE2K-mediated ubiquitination pattern of a specific substrate similar to the one we observe in our *in-vitro* ubiquitination assays for H3. The authors described this as polyubiquitination events. Similar to our results, polyubiquitinated bands are below 100 kDa and appear as distinct bands rather than smeared bands.

Notably, we found that UBE2K induces the formation of a ubiquitinated form of H3 of approximately 48 kDa. According to the molecular weight, this band could correspond to H3 tagged with a chain of four ubiquitins, which is a mark for proteasomal recognition and degradation. However, Reviewer #1 argues that this signal and others below can be caused by multi-monoubiquitination of histone H3. Here it is important to remark that our high-coverage sequencing of purified histones based on filter-aided sample preparation (FASP)-proteomics only revealed two potential ubiquitination sites in histone H3 (Supplementary Fig. 12). We agree that we cannot discard that the ubiquitinated H3 signal at approximately ~32 kDa may be caused by monoubiquitination events at two different lysine sites. However, this is a less likely explanation for the ubiquitinated H3 bands at higher molecular weights such as ~48 kDa, unless there are additional ubiquitination sites that we could not detect by FASP-proteomics. Reviewer #1 now suggests in the second revision to perform experiments with mutated ubiquitin variants that cannot form polyubiquitination in the *in-vitro* assay. To reach solid conclusions from these experiments, we would need to assess multiple single and double ubiquitin variants in the different lysine sites that can form polyubiquitin chains (e.g, K11, K48 and K67 just to mention the ones primarily associated with protein degradation). We believe that this is an interesting follow-up of our work to determine the specific polyubiquitin linkages, but it is out of the scope of this manuscript.

Nevertheless, we agree that our *in-vitro* experiments alone cannot discard that the ubiquitinated H3 bands might be caused by multi-monoubiquitination in this assay. We have now discussed this point in the Discussion section. However, we have presented additional and more relevant experiments that support polyubiquitination of H3 induced by UBE2K in cells. For instance, we performed Re-IP experiments of histone H3 as suggested by Reviewer #1 in his/her previous comments. These experiments were followed by western blot with antibody against polyubiquitinated proteins and demonstrated that UBE2K overexpression increases polyubiquitination of H3 in human cells (Figure 9i). Moreover, by using an antibody against ubiquitin to detect all the forms of ubiquitinated H3, we could confirm polyubiquitination of H3. However, we could not detect monoubiquitinated H3 in control or UBE2K overexpression conditions, suggesting that UBE2K mainly modulates H3 polyubiquitination in human cells. In our first revision, we already presented data with anti-ubiquitin antibody from the first IP (Supplementary Figure 14a). In addition, we have now included data with anti-ubiquitin antibody from Re-IP experiments to further support our conclusions (Supplementary Figure 15). Although in a less direct manner, the experiments with PSMD11 shRNA are also supportive of UBE2K-mediated polyubiquitination of H3 (Figures 9g, h). Importantly, knockdown of PSMD11, a key regulator of 26S proteasomes that degrade polyubiquitinated proteins, blocked the degradation of H3 induced by UBE2K overexpression, further supporting a role of the 26S proteasome in this process.

2. I felt that the results in previous Fig. 6d could not support their claim for a

polyubiquitin-dependent degradation of H3, and suggested that the authors should perform an in-vitro degradation assay with the different types of proteasomes. Instead, the authors addressed this issue using shRNA to knockdown the related proteasome activators. Because the 19S subunit PSMD11 is essential for cell viability, the effects of PSMD11 knockdown on H3 levels could be indirectly caused by the UBE2K-mediated monoubiquitination of other substrates. Moreover, the quality of the images in the revised Fig. 9g and 9 h is too poor to reflect the difference. For instance, the band for shRNA of PSMD4/PA200 (Fig. 9g) or PSMD11 (Fig. 9h) is much broader than the control band. Thus, the authors should perform an in-vitro degradation assay with the different types of proteasomes, if they prefer to claim that the degradation of H3 is catalyzed by the 26S proteasome, instead of the PA200-proteasome. Otherwise, this conclusion should be revised into that H3 is degraded by proteasomes in general, instead of the 26S proteasome specifically.

We agreed with Reviewer #1 regarding that previous Fig. 6d of our first submission alone could not support polyubiquitin-dependent degradation of H3, because MG132 can inhibit proteasomal degradation of all substrates and this effect could be indirect. To address this concern, we explained in our rebuttal letter the numerous reasons why we decided to focus on experiments in cells rather than in-vitro degradation assays. For instance, this experiment would require first an in-vitro ubiquitination assay of H3 followed by in-vitro degradation assays using distinct types of purified proteasomes, which are inherently labile and, thus, their activity and assembly can be easily affected by the conditions of the ubiquitination assay. Because the physiological substrates of 26S proteasomes are usually polyubiquitinated proteins, we should compare the ability of distinct isolated proteasomes to reduce the levels of ubiquitinated H3 after the in-vitro ubiquitination assay. However, only part of the recombinant H3 is ubiquitinated in the in-vitro assay and it would be challenging to conclude whether a potential decrease in the levels of polyubiquitinated H3 is due to the deubiquitination activity of 19S components or proteolytic activity. Thus, the suggested in-vitro degradation assays are not only technically challenging, but might also lack the physiological content. Therefore, it appears out of the scope of this work to address the activities of different proteasomal complexes in vitro, which might not necessarily reflect the in vivo regulation of histone H3 degradation. In this regard, experiments in cells was an excellent suggestion by Reviewer #1 and we are very grateful for it because it led to exciting results that strengthened our conclusions. It is important to note again that immunoprecipitation experiments of H3 demonstrated that UBE2K overexpression increases polyubiquitination of H3 in human cells (Figure 9i). In fact, UBE2K-mediated polyubiquitination of H3 in cells is more robust than the changes observed in in-vitro experiments, supporting our decision to focus on cell systems. In addition, overexpression of UBE2K decreases H3 levels in cells (Figures 9d and 9h). Thus, cell models provide a bona fide system to study the link between UBE2K with polyubiquitination and proteasomal degradation of H3. We respectfully disagree with Reviewer #1's comment regarding the quality of images in Fig. 9f and Fig. 9h. In fact, the differences between PSMD11 shRNA and PSME4 shRNA are already quite striking in the images presented to support our conclusions regarding a

prominent role of 26S proteasomes in H3 degradation (Fig. 9g). The images in Fig. 9h also strongly support that PSMD11 shRNA reduces the degradation of H3 induced by UBE2K overexpression. Nevertheless, we also presented quantifications (normalized by B-actin) of several experiments. These quantitative data further supported our conclusions:

- 1) A moderate knockdown of PSMD11 (approx. 40%) was sufficient to induce a slight (24%), but significant decrease in proteasome activity (**Fig. 9e, f**). PSME4 knockdown induced a similar decrease in proteasome activity (**Fig. 9e, f**).
- 2) Even when PSMD11 and PSME4 knockdown induced a similar decrease in total proteasome activity (**Fig. 9e, f**), PSMD11 shRNA resulted in a stronger and significant increase in histone H3 levels when compared to PSME4 shRNA (**Fig. 9g**). Thus, these data indicate that the 26S proteasome has a more relevant role in the degradation of H3 in comparison with other proteasome assemblies.
- 3) PSMD11 knockdown diminished the degradation of H3 induced by UBE2K overexpression (**Fig. 9h**), supporting an important role of 26S proteasome in UBE2K-mediated degradation of H3.

To explain our results, the Reviewer raises a complex and intricate, but fascinating hypothesis: "Because the 19S subunit PSMD11 is essential for cell viability, the effects of PSMD11 knockdown on H3 levels could be indirectly caused by the UBE2K-mediated monoubiquitination of other substrates". This hypothesis could even reveal a prominent role of UBE2K-mediated monoubiquitination in the effects triggered by proteasome inhibition and cell death, urging to revisit the conclusions of many manuscripts where specific proteasome targets were identified using proteasome inhibitors. However, we believe that there is no evidence in the literature or from our data to support this hypothesis. In fact, experimental data support our conclusion that UBE2K has direct effects on ubiquitination and proteasomal degradation of H3:

- 1) UBE2K is a Class II E2 that preferentially synthesizes Lys48-linked chains on monoubiquitylated substrates, or forms Lys48-linked diubiquitin in the absence of an E3 (Middleton and Day, 2015). Thus, the process of monoubiquitylation might not come into play in cells.
- 2) The mild knockdown of PSMD11 that we used did not have any strong effects on cell viability. As most of the proteasome subunits, PSMD11 is essential for cell viability because it is required for assembly and activity of the proteasome, which eventually determines cell viability. Thus, strong knockdown levels can promote cell death similar to proteasome inhibitor treatment during extended periods. For this reason, we induced a mild knockdown of PSMD11, which allowed us to generate stable PSMD11 shRNA cells that can replicate continuously and do not exhibit obvious phenotypes in cell morphology, viability or proliferation rates when compared with control non-targeting shRNA stable cells. This mild knockdown was sufficient to induce a significant decrease in proteasome activity (approx. 24%), but much lower than proteasome inhibitor

treatment (fig. 9f). We apologize because we did not explain this in detail in our first revision, leading to a misunderstanding. We have now made this point more clear in the main text.

- 3) *We find that UBE2K interacts with H3. Our in vitro assays demonstrate that UBE2K can regulate ubiquitination of H3. Our IP experiments show that UBE2K induces polyubiquitination of H3 in cells. Both proteasome inhibitor and PSMD11 shRNA diminish the degradation of UBE2K induced by UBE2K overexpression.*

Therefore, we believe that experiments in cells based on knockdown of activators of distinct proteasome assemblies support our conclusions. These assays are more relevant and informative for our conclusions than in-vitro degradation assays. As suggested by Reviewer #1, we have now re-written the text to note that whereas our results indicate that 26S proteasomes have a prominent role in the degradation of H3, we cannot discard that other proteasome assemblies are also involved in this process. For instance, we discuss that we cannot discard that a stronger knockdown of PSME4 may induce a further reduction in proteasome activity, which eventually could lead to significant changes in histone H3 levels.

Reviewer #2 (Remarks to the Author):

My concerns have been adequately addressed and I support publication in Communications Biology.

We thank Reviewer #2 for his insightful comments that improved our manuscript and for supporting its publication in Communications Biology.

Reviewer #3 (Remarks to the Author):

The author addressed fully the questions I raised in last version and now they offered more evidences to support their conclusion. It seems that UBE2K has impact on both protein level of H3 itself and H3K9Me3 modification through ubiquitination and H3K9me3/SETDB1 axis, respectively. However, it is not clear which, ubiquitination-mediated stability of H3 or H3K9Me3, is the key player in regulating the neurogenic potential of the hESCs. Second, what is the relation between H3K9Me3 and UBE2K-mediated ubiquitination of H3. The authors should discuss these points.

Although our results indicate a role of UBE2K-mediated changes of H3K9me3 in neuronal differentiation, we completely agree with Reviewer #3 that it still not clear whether H3K9 trimethylation or H3 stability is the key player in regulating the neurogenic potential of hESCs. For instance, dysregulated H3 levels may also contribute to diminishing neurogenic potential and even has a more prominent role in this phenotype relative to H3K9me3 changes. In addition, we found that whereas

UBE2K knockdown did not significantly change the ratio of H3K4me3, H3K27me3, H3K27ac when normalized to total H3, it increased overall levels of these modifications. Thus, we cannot discard that upregulated H3K4me3, H3K27me3, H3K27ac amounts resulting from increased H3 levels also have a key role in regulating neurogenic potential. Moreover, as indicated by Reviewer #3, we do not know whether UBE2K-mediated ubiquitination and stability of H3 could modulate H3K9 trimethylation. Our results indicate that UBE2K knockdown impairs HTT and SETDB1 levels. Since these proteins form complexes with UBE2K and H3, an intriguing hypothesis is that UBE2K-mediate stability of H3 determines changes in the levels or interactions of key regulators of H3K9me3. We have now discussed this thoroughly in the Discussion section.